# Efficient Diffusion Models: A Survey

**Hui Shen**[1,†]                                              *shen.1780@osu.edu*

**Jingxuan Zhang**[2,†]                                        *jz97@iu.edu*

**Boning Xiong**[3,†]                                          *bnxiong24@m.fudan.edu.cn*

**Rui Hu**[4,†]                                                *marshallrui@gmail.com*

**Shoufa Chen**[5]                                            *shoufachen66@gmail.com*

**Zhongwei Wan**[1]                                            *wan.512@osu.edu*

**Xin Wang**[1]                                                *wang.15980@osu.edu*

**Yu Zhang**[6]                                                *zhangyu.ansel@gmail.com*

**Zixuan Gong**[6]                                            *gongzx@tongji.edu.cn*

**Guangyin Bao**[6]                                            *baogy@tongji.edu.cn*

**Chaofan Tao**[5]                                            *tcftrees@gmail.com*

**Yongfeng Huang**[7]                                          *1155187959@link.cuhk.edu.hk*

**Ye Yuan**[8]                                                *yuanye_pku@pku.edu.cn*

**Mi Zhang**[1]                                                *mizhang.1@osu.edu*[*]

[1] *The Ohio State University*   [2] *Indiana University*   [3] *Fudan University*   [4] *Hangzhou City University*
[5] *The University of Hong Kong*   [6] *Tongji University*   [7] *The Chinese University of Hong Kong*   [8] *Peking University*

**Reviewed on OpenReview:** *https://openreview.net/forum?id=wHECkBOwyt*

## Abstract

Diffusion models have emerged as powerful generative models capable of producing high-quality contents such as images, videos, audio, and text, demonstrating their potential to revolutionize digital content generation. However, these capabilities come at the cost of their significant resource demands and lengthy generation time, underscoring the need to develop efficient techniques for practical deployment. In this survey, we provide a systematic and comprehensive review of research on efficient diffusion models. We organize the literature in a taxonomy consisting of three main categories, covering distinct yet interconnected efficient diffusion model topics from algorithm-level, system-level, and framework perspective, respectively. We have also created a GitHub repository where we organize the papers featured in this survey at https://github.com/AIoT-MLSys-Lab/Efficient-Diffusion-Model-Survey. We hope our survey can serve as a valuable resource to help researchers and practitioners gain a systematic understanding of efficient diffusion model research and inspire them to contribute to this important and exciting field.

---

[*]The marker [†] denotes Co-first authors.

# 1  Introduction

Diffusion models kickstart a new era in the field of artificial intelligence generative content (AIGC), garnering unprecedented attention (Yang et al., 2023b; Croitoru et al., 2023). Especially in the context of image synthesis tasks, diffusion models have demonstrated impressive and diverse generative capabilities. The powerful cross-modal capabilities of diffusion models have also further fueled the vigorous development of downstream tasks (Chen et al., 2023b). Despite the increasing maturity of diffusion model variants after numerous iterations (Zhang et al., 2023d; Xu et al., 2023a), generating high-resolution complex natural scenes remains both time-consuming and computationally intensive, whether the initial pixel-level approach (Ho et al., 2020) or the latent space variant (Rombach et al., 2022). Therefore, in order to optimize user-level deployment of diffusion models, researchers have never ceased their pursuit of efficient diffusion models.

Despite the growing popularity of diffusion models in recent years, one of the significant issues with diffusion model is that its multi-step denoising procedure requires numerous timesteps to reconstruct a high-quality sample from noise. This multi-step process is not only time-consuming but also computationally intensive, resulting in a heavy workload. Therefore, improving the efficiency of diffusion models is crucial. In recent years, various studies have been presented to address this problem, such as controlling the noise added during training (Hang & Gu, 2024; Chen et al., 2023a) and selecting appropriate sampling timesteps (Watson et al., 2021; Sabour et al., 2024), among others.

While there are numerous comprehensive surveys on diffusion models (Yang et al., 2023b; Chen et al., 2024; Croitoru et al., 2023; Cao et al., 2024) and those focused on specific fields and tasks (Ulhaq et al., 2022; Lin et al., 2024c; Kazerouni et al., 2023; Lin et al., 2024b; Peng et al., 2024b; Daras et al., 2024), discussions on the efficiency of diffusion models are notably scarce. The only existing survey addressing efficiency (Ma et al., 2024c) serves as an initial exploration in this area. In our work, we provide a more comprehensive and detailed taxonomy of efficient techniques, covering a broader and more recent collection of research papers.

The overarching goal of this survey is to provide a holistic view of the technological advances in efficient diffusion models from **algorithm-level**, **system-level**, and **framework** perspectives, as illustrated in Figure 1. These four categories cover distinct yet interconnected research topics, collectively providing a systematic and comprehensive review of efficient diffusion models research. Specifically,

- **Algorithm-Level Methods:** Algorithm-level methods are critical for improving the computational efficiency and scalability of diffusion models, as their training and inference processes are often resource-intensive. In §3, we survey efficient techniques that cover research directions related to efficient training, efficient fine-tuning, efficient sampling, and model compression.

- **System-Level Methods:** System-level methods aim to optimize the infrastructure and computational resources required for training and deploying diffusion models. In §4, we survey efficient techniques that cover research directions related to optimized hardware-software co-design, parallel computing, and caching techniques.

- **Frameworks:** The advent of diffusion models necessitates the development of specialized frameworks to efficiently handle their training, fine-tuning, inference, and serving. While mainstream AI frameworks such as TensorFlow and PyTorch provide the foundations, they lack built-in support for specific optimizations and features crucial for diffusion models. In §5, we survey existing frameworks specifically designed for efficient diffusion models, covering their unique features, underlying libraries, and specializations.

In addition to the survey, we have established a GitHub repository where we compile the papers featured in this survey at https://github.com/AIoT-MLSys-Lab/Efficient-Diffusion-Model-Survey. We will actively maintain it and incorporate new research as it emerges.

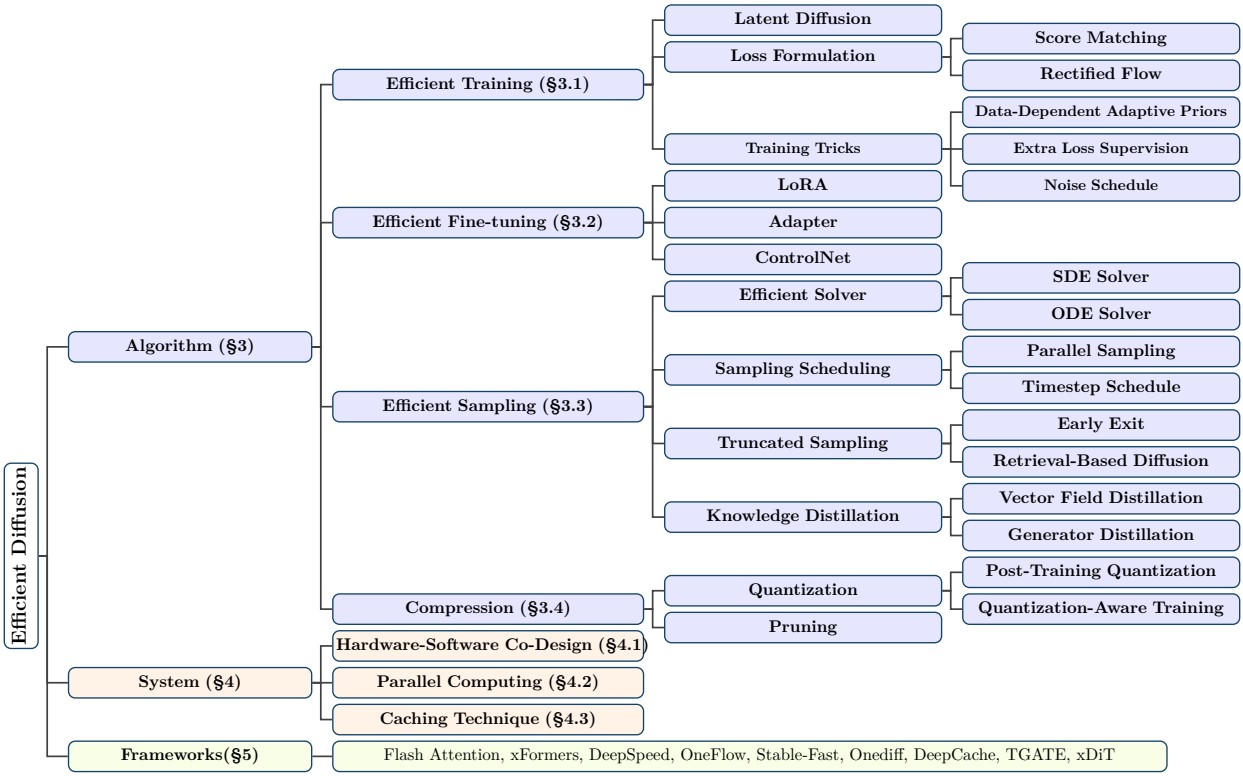

Figure 1: Taxonomy of efficient diffusion model literature.

## 2 Background and Applications

### 2.1 Basic formulas for Diffusion models

Diffusion models have emerged as powerful generative models capable of producing high-quality samples across various domains. This section explores the theoretical foundations and recent advancements in the diffusion model framework. We begin by examining the fundamental formulation of Denoising Diffusion Probabilistic Models (DDPMs) (Ho et al., 2020), which leverage forward and backward stochastic processes to gradually transform data into noise and then reverse this process for generation. We then explore Score Matching (Hyvärinen & Dayan, 2005) as an alternative formulation that directly optimizes gradient fields of probability densities. Continuing our discussion, we analyze how Stochastic Differential Equations (SDEs) and Ordinary Differential Equations (ODEs) (Song et al., 2020c) provide continuous perspectives on diffusion. Finally, we introduce Flow Matching (Lipman et al., 2022) as a recent paradigm that offers improved efficiency by directly learning vector fields that transform distributions, utilizing ODEs to provide a deterministic process.

#### 2.1.1 DDPM

To better understand the directions for improving efficient diffusion models, it is essential first to comprehend the fundamental framework of diffusion models. Denoising Diffusion Probabilistic Models (DDPMs) (Ho et al., 2020) generate data through a process analogous to thermodynamic diffusion, consisting of two key components: a forward process and a reverse process. These processes work in concert to enable high-quality generative modeling.

The forward process in DDPM is a fixed Markov chain involving gradually adding Gaussian noise to the data until it becomes pure noise. $q(\mathbf{x}_0)$ is denoted as the true data distribution, and assuming that $\mathbf{x}_0 \sim q(\mathbf{x}_0)$ represents sampled data from this distribution. The forward process noted as $q(\mathbf{x}_{1:T}|\mathbf{x}_0)$, adds Gaussian

noise step by step, transforming the data from $\mathbf{x}_0$ to $\mathbf{x}_T$:

$$q(\mathbf{x}_{1:T}|\mathbf{x}_0) := \prod_{t=1}^{T} q(\mathbf{x}_t|\mathbf{x}_{t-1}), \quad q(\mathbf{x}_t|\mathbf{x}_{t-1}) := \mathcal{N}(\mathbf{x}_t; \sqrt{\alpha_t}\mathbf{x}_{t-1}, \beta_t I) \tag{1}$$

$\beta_t$ is defined as the variance of the noise added at each timestep. We then convert this to $\alpha_t = 1 - \beta_t$. Additionally, $\bar{\alpha}_t = \prod_{s=1}^{t} \alpha_s$ is defined as the cumulative product of $\alpha_t$, following the formulation by Sohl-Dickstein et al. (Sohl-Dickstein et al., 2015). This cumulative product allows for modeling the transition from the original data $\mathbf{x}_0$ to $\mathbf{x}_t$ as a Gaussian distribution:

$$q(\mathbf{x}_t|\mathbf{x}_0) = \mathcal{N}(\mathbf{x}_t; \sqrt{\bar{\alpha}_t}\mathbf{x}_0, (1 - \bar{\alpha}_t)\mathbf{I}) \tag{2}$$

This expression describes the distribution of $\mathbf{x}_t$ given the initial data $\mathbf{x}_0$. It indicates that $\mathbf{x}_t$ can be expressed as a linear combination of $\mathbf{x}_0$ and noise, where $\boldsymbol{\epsilon} \sim \mathcal{N}(\mathbf{0}, \mathbf{I})$ represents standard Gaussian noise:

$$\mathbf{x}_t = \sqrt{\bar{\alpha}_t}\mathbf{x}_0 + \sqrt{1 - \bar{\alpha}_t}\boldsymbol{\epsilon} \tag{3}$$

The reverse process, in contrast, aims to gradually denoise and reconstruct the original data by reversing the noise addition performed in the forward process. This reverse process is modeled as a Markov chain where each step transitions from $\mathbf{x}_t$ to $\mathbf{x}_{t-1}$ using a learned conditional probability distribution $p_\theta(\mathbf{x}_{t-1}|\mathbf{x}_t)$. The overall process is expressed as:

$$p_\theta(\mathbf{x}_{0:T}) := p(\mathbf{x}_T) \prod_{t=1}^{T} p_\theta(\mathbf{x}_{t-1}|\mathbf{x}_t), \quad p_\theta(\mathbf{x}_{t-1}|\mathbf{x}_t) := \mathcal{N}(\mathbf{x}_{t-1}; \mu_\theta(\mathbf{x}_t, t), \Sigma_\theta(\mathbf{x}_t, t)) \tag{4}$$

where $p(\mathbf{x}_T)$ is the initial Gaussian distribution at the final time step $T$, and $p_\theta(\mathbf{x}_{t-1}|\mathbf{x}_t)$ represents the conditional probability distribution learned by the model to transition between states. The mean $\mu_\theta(\mathbf{x}_t, t)$ and covariance $\Sigma_\theta(\mathbf{x}_t, t)$ are parameterized functions of the state $\mathbf{x}_t$, the time step $t$, and the model parameters $\theta$. In the training process, the optimization objective is to minimize the negative log-likelihood using the variational bound to approximate the true data distribution:

$$\mathbb{E}[-\log p_\theta(\mathbf{x}_0)] \leq \mathbb{E}_q\left[-\log \frac{p_\theta(\mathbf{x}_{0:T})}{q(\mathbf{x}_{1:T}|\mathbf{x}_0)}\right] = \mathbb{E}_q\left[-\log p(\mathbf{x}_T) - \sum_{t\geq 1} \log \frac{p_\theta(\mathbf{x}_{t-1}|\mathbf{x}_t)}{q(\mathbf{x}_t|\mathbf{x}_{t-1})}\right] =: L \tag{5}$$

This objective function decomposes the optimization problem into KL divergences for each timestep, progressively optimizing the reverse process. Expanding the KL terms and using the conditional Gaussian form evaluates the difference between the forward and reverse processes, ultimately simplifying the process into a mean squared error form:

$$L_{\text{simple}}(\theta) := \mathbb{E}_{t,\mathbf{x_0},\boldsymbol{\epsilon}}\left[\|\boldsymbol{\epsilon} - \boldsymbol{\epsilon}_\theta(\sqrt{\bar{\alpha}_t}\mathbf{x}_0 + \sqrt{1 - \bar{\alpha}_t}\boldsymbol{\epsilon}, t)\|^2\right] \tag{6}$$

### 2.1.2 Score Matching

Score matching, introduced by Hyvärinen & Dayan (2005), serves as an effective approach for estimating unnormalized models by minimizing the Fisher divergence between the score function of data distribution $\boldsymbol{s}_d(\mathbf{x}) = \nabla_\mathbf{x} \log p_d(\mathbf{x})$ and the score function of model distribution $\boldsymbol{s}_m(\mathbf{x}; \boldsymbol{\theta}) = \nabla_\mathbf{x} \log p_m(\mathbf{x}; \boldsymbol{\theta})$. This approach avoids the need to compute the intractable partition function $\boldsymbol{Z_\theta}$, a common problem in Maximum Likelihood Estimation (MLE).

While DDPM directly optimizes the noise prediction in Eq.(6), score matching objectives can directly be estimated on a dataset and optimized with stochastic gradient descent. The loss function for score matching takes a different approach, formulated as follows:

$$L(\boldsymbol{\theta}) = \frac{1}{2}\mathbb{E}_{p_d(\mathbf{x})}\left[\|\boldsymbol{s}_m(\mathbf{x}; \boldsymbol{\theta}) - \boldsymbol{s}_d(\mathbf{x})\|^2\right] \tag{7}$$

Since it typically does not have access to the true score function of the data $\boldsymbol{s}_d(\mathbf{x})$, Hyvärinen & Dayan (2005) introduced integration by parts as $L(\boldsymbol{\theta}) = J(\boldsymbol{\theta}) + C$ to derive an alternative expression that does not require direct access to $\mathbf{x}_d(\mathbf{x})$:

$$J(\boldsymbol{\theta}) = \mathbb{E}_{p_d(\mathbf{x})}\left[\mathrm{tr}(\nabla_{\mathbf{x}}\boldsymbol{s}_m(\mathbf{x};\boldsymbol{\theta})) + \frac{1}{2}\|\boldsymbol{s}_m(\mathbf{x};\boldsymbol{\theta})\|^2\right] \tag{8}$$

In this expression, $\mathrm{tr}(\cdot)$ denotes the trace of the Hessian matrix of $\boldsymbol{s}_m(\mathbf{x};\boldsymbol{\theta})$. The constant $C$ is independent of $\boldsymbol{\theta}$ and can be ignored for optimization purposes. The final form of the unbiased estimator used for training is:

$$\hat{J}(\boldsymbol{\theta};\mathbf{x}_1^N) = \frac{1}{N}\sum_{i=1}^{N}\left[\mathrm{tr}(\nabla_{\mathbf{x}}\boldsymbol{s}_m(\mathbf{x}_i;\boldsymbol{\theta})) + \frac{1}{2}\|\boldsymbol{s}_m(\mathbf{x}_i;\boldsymbol{\theta})\|^2\right] \tag{9}$$

### 2.1.3 Solvers

Given that the cost of sampling escalates proportionally with the number of discretized time steps, many researchers have concentrated on devising discretization schemes that reduce the number of time steps. A key insight emerges from reexamining the discrete forward process in the original DDPM formulation Eq.(1), as we reduce the step size between consecutive steps, the process naturally approaches a continuous transformation. Consequently, adopting learning-free methods using SDE or ODE solvers (Song et al., 2020c) has been proposed.

**SDE Solver.** Song et al. (2020c) firstly presents a stochastic differential equation (SDE) that smoothly transforms a complex data distribution to a known prior distribution by slowly injecting noise and a corresponding reverse-time SDE that transforms the prior distribution back into the data distribution by slowly removing the noise. The discrete noise addition steps in Eq.(1) are reformulated into a continuous process:

SDE accomplishes the transformation from data to noise in the diffusion training process through the following equation:

$$\mathrm{d}\mathbf{x} = \mathbf{f}(\mathbf{x}, t)\,\mathrm{d}t + g(t)\,\mathrm{d}\bar{\mathbf{w}} \tag{10}$$

where $\bar{\mathbf{w}}$ denotes the standard Wiener process, also known as Brownian motion. $\mathbf{f}(\mathbf{x}, t)$ is a vector-valued function called the drift coefficient of $\mathbf{x}(t)$, and $g(t)$ is a scalar function.

Similarly, the reverse process Eq.(4) can be generalized to a continuous-time formulation:

$$\mathrm{d}\mathbf{x} = \left[\mathbf{f}(\mathbf{x}, t) - g(t)^2\nabla_{\mathbf{x}}\log q_t(\mathbf{x})\right]\mathrm{d}t + g(t)\mathrm{d}\bar{\mathbf{w}} \tag{11}$$

$\bar{\mathbf{w}}$ is a standard Wiener process when time flows backward from $T$ to 0, $dt$ is an infinitesimal negative timestep and $\nabla_{\mathbf{x}}\log q_t(\mathbf{x})$ represent the score function that we mentioned in Eq.(10). In the diffusion process, reverse-time SDE converts noise into data gradually. The complete SDE process is shown in Figure 2.

**ODE Solver.** Unlike SDE solvers, the trajectories generated by ordinary differential equation (ODE) solvers are deterministic (Song et al., 2020c), remaining unaffected by stochastic variations. Consequently, these

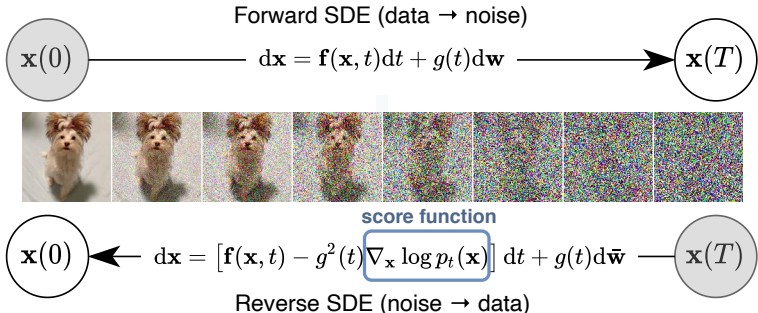

Figure 2: Overview of forward SDE process and reverse SDE process (Song et al., 2020c).

deterministic ODE solvers tend to achieve convergence more rapidly compared to their stochastic counterparts, although this often comes at the expense of a marginal reduction in sample quality. The corresponding deterministic process Eq.(12) can be derived from the reverse-time SDE Eq.(11) by removing the stochastic term $g(t)\mathrm{d}\bar{\mathbf{w}}$, resulting in a deterministic process that shares the same marginal probability densities as the reverse-SDE:

$$\mathrm{d}\mathbf{x} = \left[\mathbf{f}(\mathbf{x}, t) - \frac{1}{2}g(t)^2 \nabla_{\mathbf{x}} \log q_t(\mathbf{x})\right] \mathrm{d}t \tag{12}$$

The forward process also exhibits a similar distinction between SDE and ODE approaches, yielding a deterministic process that preserves the same marginal distributions:

$$\mathrm{d}\mathbf{x} = \mathbf{f}(\mathbf{x}, t)\mathrm{d}t \tag{13}$$

### 2.1.4 Flow Matching

Flow Matching (FM) (Lipman et al., 2022) is a new paradigm for generative modeling based on Continuous Normalizing Flows (CNFs). This approach allows us to train CNFs at an unprecedented scale and offers greater efficiency compared to traditional diffusion models.

To better understand the process of flow matching, we need to dive into the preliminaries. First of all, we make $\mathbb{R}^d$ denote the data space with data points $x = (x^1, \ldots, x^d) \in \mathbb{R}^d$. Then, we need to understand several key concepts: Probability density path $p_t : [0, 1] \times \mathbb{R}^d \to \mathbb{R}_{>0}$, which is a time-dependent probability density function, and Time-dependent vector field $v_t : [0, 1] \times \mathbb{R}^d \to \mathbb{R}^d$, which describes how data points change over time. A vector field $v_t$ can be used to construct a time-dependent diffeomorphic map, defined by an ordinary differential equation (ODE):

$$\frac{\mathrm{d}}{\mathrm{d}t}\phi_t(x) = v_t(\phi_t(x)) \tag{14}$$

$$\phi_0(x) = x \tag{15}$$

A Continuous Normalizing Flow (CNF) is a generative model that parametrizes the time-dependent vector field $v_t(x; \theta)$ using a neural network (Chen et al., 2018), where $\theta$ represents the learnable parameters. This vector field defines a flow $\phi_t$ that transforms the probability distribution from a simple prior density $p_0$ to a more complex target density $p_1$ through the push-forward equation. The core of CNF is using this flow to reshape probability distributions:

$$p_t = [\phi_t]_* p_0 \tag{16}$$

The push-forward operator $*$ is defined as:

$$[\phi_t]_* p_0(x) = p_0(\phi_t^{-1}(x)) \det\left[\frac{\partial \phi_t^{-1}}{\partial x}(x)\right] \tag{17}$$

The FM loss is formally defined as:

$$\mathcal{L}_{\mathrm{FM}}(\theta) = \mathbb{E}_{t \sim \mathcal{U}[0,1], \, x \sim p_t(x)} \|v_t(x) - u_t(x)\|^2, \tag{18}$$

where: $u_t(x)$ generates a predefined probability density path $p_t(x)$, $p_0(x) = \mathcal{N}(x|0, I)$ is a simple prior distribution (e.g., Gaussian noise), and $p_1(x) \approx q(x)$ approximates the data distribution.

While FM is conceptually straightforward, directly optimizing this objective is intractable due to the lack of closed-form expressions for $p_t$ and $u_t$. To address this, FM constructs conditional probability paths $p_t(x|x_1)$ and vector fields $u_t(x|x_1)$ per data sample $x_1 \sim q(x_1)$, where $p_0(x|x_1) = p(x)$ at time $t = 0$, and $p_1(x|x_1) = \mathcal{N}(x|x_1, \sigma^2 I)$ at $t = 1$ (a normal distribution with $x_1$ mean and a sufficiently small standard deviation $\sigma > 0$). These conditional paths are then aggregated into global counterparts through marginalization:

$$p_t(x) = \int p_t(x|x_1)q(x_1)\mathrm{d}x_1 \tag{19}$$

This approach avoids explicit modeling of the intractable marginal distributions and enables scalable training, while ensuring that at $t = 1$, the marginal distribution $p_1(x)$ approximates the data distribution $q(x)$.

The marginal vector field can then be defined as:

$$u_t(x) = \int u_t(x|x_1) \frac{p_t(x|x_1)q(x_1)}{p_t(x)} \mathrm{d}x_1 \tag{20}$$

A key theorem in the paper shows that this constructed marginal vector field $u_t$ in Eq.(20) indeed generates the marginal probability path $p_t$ in Eq.(19).

However, computing the FM objective is still infeasible because it involves complex integrals. To address this issue, the paper proposes the Conditional Flow Matching (CFM) objective:

$$\mathcal{L}_{\mathrm{CFM}}(\theta) = \mathbb{E}_{t,q(x_1),p_t(x|x_1)} \|v_t(x) - u_t(x|x_1)\|^2, \tag{21}$$

The second key theorem in the paper proves that the FM and CFM objectives have identical gradients with respect to $\theta$, so optimizing CFM is equivalent to optimizing FM, but computationally simpler. Compared to score matching, flow matching is faster because score matching typically requires computing Hessian matrices in Eq(9), while CFM cleverly avoids such complex calculations by only computing the L2 distance between vector fields. In conclusion, it's worth noting that Flow Matching can be theoretically connected with other diffusion model formulations. As summarized by Gao et al. (2024), flow matching, score-based models, and traditional diffusion models can be understood within a unified mathematical framework. These approaches essentially represent different parameterizations of the same continuous-time process. While traditional diffusion models like DDPM (Ho et al., 2020) learn to estimate noise or score functions, and score-based models directly optimize gradient fields (Hyvärinen & Dayan, 2005), flow matching learns the vector field itself, directly capturing the transformation map between distributions.

## 2.2  Applications

Building on the foundational principles of diffusion models outlined in §2.1, this section surveys their practical deployment across diverse generative tasks, with a specific emphasis on efficiency-driven innovations. As computational demands and real-world applicability become increasingly critical, researchers have adapted diffusion models to optimize resource usage while preserving or enhancing output quality. Here, we explore key application domains—image, video, text, audio, and 3D generation—highlighting techniques that reduce inference time, memory footprint, and training complexity. These advancements underscore the adaptability of diffusion models and their growing impact in addressing the challenges of scalable, high-fidelity content generation.

### 2.2.1  Image Generation

Image generation is the primary application domain for efficient diffusion models. Researchers have been developing various approaches to optimize both computational resources and generation quality. The efficiency improvements in this field are well exemplified by several influential works. For example, Stable Diffusion (Rombach et al., 2022) pioneered the concept of efficient image generation by operating in a compressed latent space rather than pixel space, significantly reducing memory and computational requirements while maintaining high-quality outputs. Latent Consistency Models (LCM) (Luo et al., 2023a) further pushed the boundaries by enabling high-quality image generation in just 4 steps through careful design of the consistency loss and distillation process. Progressive distillation (Salimans & Ho, 2022) demonstrated that through a student-teacher framework, diffusion models could achieve comparable quality to 50-step sampling using only 2-8 inference steps. ControlNet (Zhang et al., 2023d) introduced an efficient architecture for adding spatial conditioning controls to pretrained diffusion models through zero-initialized convolutions, enabling diverse control capabilities without compromising model efficiency. More recently, Efficient Diffusion (EDM) (Karras et al., 2022) presented a comprehensive framework for training and sampling diffusion models more efficiently, introducing improvements in both training stability and inference speed while maintaining state-of-the-art generation quality.

### 2.2.2 Video Generation

Following the rapid escalation in image generation, video generation similarly garnered widespread attention (Melnik et al., 2024; Ho et al., 2022b; Xing et al., 2023). The heavy model size and the substantial computational costs have further intensified the focus on developing more efficient methods for video generation (Zhang et al., 2023b; Liu et al., 2023b; Xing et al., 2024; Wang et al., 2023; Lee et al., 2024b). For example, Zhang et al. (2023b) introduced AdaDiff, a lightweight framework designed to optimize a specialized policy gradient method tailored to individual text prompts. This approach facilitates the design of reward functions and enables an effective trade-off between inference time and generation quality. Specifically to the training process, Liu et al. (2023b) proposed an efficient training framework ED-T2V to freeze the pretraining model (Rombach et al., 2022) training additional temporal modules. Similarly, Xing et al. (2024) suggested using spatial and temporal adapters. In their approach, the original T2I model remains frozen during training, and only the newly added adapter modules are updated. Unlike the works above, Wang et al. (2023) presented VideoLCM, incorporating consistency distillation in the latent space. VideoLCM efficiently distills knowledge from a pretraining model, maintaining fidelity and temporal coherence while improving inference speed. Lee et al. (2024b) introduces a grid diffusion model by representing a video as a grid of images. It employs key grid image generation and autoregressive grid interpolation to maintain temporal consistency. Moreover, Ceylan et al. (2023) leverages self-attention feature injection and guided latent updates, efficiently repurposing image models for video editing, enabling high-quality, consistent edits at minimal computational overhead. Yin et al. (2023) proposes NUMA-XL, using a hierarchical coarse-to-fine approach to enable parallelizable, exponential scaling of video length. It achieves a great reduction in inference time and is trained directly on long sequences, ensuring long-term consistency.

### 2.2.3 Text Generation

Efficient diffusion models offer a fresh perspective in text generation through their stochastic and iterative processes. However, they encounter several challenges when applied to discrete data types such as text. For instance, the common use of Gaussian noise is less effective for discrete corruption, and the objectives designed for continuous spaces become unstable in the text diffusion process, particularly at higher dimensions. With these challenges, Chen et al. (2023a) proposed a diffusion model called Masked-Diffuse LM. In the diffusion process, a cross-entropy loss function at each diffusion step is utilized to efficiently bridge the gap between the continuous representations in the model and the discrete textual outputs. SeqDiffuSeq (Yuan et al., 2024) incorporates an encoder-decoder Transformer architecture, achieving efficient text generation through adaptive noise schedule and self-conditioning (Chen et al., 2022a) techniques. Using the same encoder-decoder architecture, Lovelace et al. (2024) presents a methodology where text is encoded into a continuous latent space. Subsequently, continuous diffusion models are employed for sampling within this space.

### 2.2.4 Audio Generation

In the field of audio generation, the application of diffusion models presents several unique challenges. First, audio data exhibits strong temporal continuity, especially in high-resolution audio generation tasks, where the model must accurately reconstruct both time-domain and frequency-domain information. Compared to images or text, even subtle distortions or noise in audio are easily perceptible by humans, directly affecting the listening experience, particularly in speech and music generation tasks. Ensuring high fidelity and maintaining the consistency of details in the generated audio is therefore crucial. Moreover, many audio generation tasks require low-latency feedback, especially in applications like speech synthesis and real-time dialogue, which makes acceleration of diffusion models essential. The multi-dimensional nature of audio data, such as time-domain, frequency-domain, stereo, and spatial audio, further complicates the generation process, requiring the model to handle these dimensions while maintaining consistency during the accelerated generation. To address these challenges, researchers have proposed various methods to accelerate diffusion models in audio generation. Some works focus on reducing the number of diffusion steps to speed up the generation process, such as Chen et al. (2020) in WaveGrad and Kong et al. (2020) in DiffWave, which optimize the network structure to reduce generation time while maintaining high audio quality. Further optimization comes from the FastDPM framework (Kong & Ping, 2021), which generalizes discrete diffusion steps to continuous ones, using a bijective mapping between noise levels to accelerate sampling without

compromising quality. FastDPM's flexibility allows it to adapt to different domains, and in the case of audio synthesis, where stochasticity plays a crucial role, it demonstrates superior performance in high-stochasticity tasks like speech generation. Through these approaches, diffusion models not only accelerate the generation process but also reduce computational costs while ensuring that audio quality remains high, meeting the demands of real-time audio generation applications.

### 2.2.5 3D Generation

As a technique closely aligned with real-world representation, 3D generation holds substantial promise across various sectors, including medical imaging, motion capture, asset production, and scene reconstruction, etc. However, when compared to 2D image generation, distinctive high-resolution elements such as volumetric data or point clouds present unique challenges, significantly escalating computational demands. Several efficient methodologies (Bieder et al., 2023; Zhou et al.; Tang et al., 2023b; Park et al., 2023; Du et al., 2024; Wu et al., 2024) have been proposed, particularly concentrating on enhancing the sampling process and optimizing the architectural framework, which further handles the computational complexity inherent. One of the most prevalent approaches involves designing more efficient sampling schedules (Bieder et al., 2023; Li et al., 2024c; Yu et al., 2024b; Zhou et al.). By utilizing larger sampling step sizes, modifying the sampling strategy between 2D and 3D, or incorporating multi-view parallelism, these techniques address the key bottlenecks in the sampling process, thereby improving sampling efficiency. Moreover, the incorporation of novel architectures, such as state-space models and lightweight feature extractors (Mo, 2024; Tang et al., 2023b), alleviates the computational burden of processing 3D data, significantly enhancing model efficiency.

Table 1: Representative applications of diffusion models.

| Task | Datasets | Metrics | Articles |
|------|----------|---------|----------|
| Image Generation | ImageNet, CIFAR, MetFace, CelebA HQ, MS COCO, UCI, FFHQ, DiffusionDB, AFHQ, LSUN, SYSTEM-X, LAION | FID, sFID, IS, NLL, MSE, CLIP Score, PSNR, LPIPS, MACs, CS, PickScore, SA, Score Matching Loss | Liu et al. (2022b), Liu et al. (2023c), Yan et al. (2024), Lee et al. (2024a), Zhu et al. (2024), etc. |
| Video Generation | MSR-VTT, InternVid, WebVid-10M, LAION, UCF-101, CGCaption, DAVIS, FlintstonesHD | FID, IS, FVD, IQS, NIQE, CLIPSIM, B-FVD-16, Pixel-MSE | Zhang et al. (2023b), Liu et al. (2023b), Xing et al. (2024), Wang et al. (2023), Lee et al. (2024b), Ceylan et al. (2023), Yin et al. (2023), etc. |
| Audio Generation | SC09, LJSpeech, Speech Commands | MOS, FID, IS, mIS, AM Score | Chen et al. (2020), Kong et al. (2020), Kong & Ping (2021), etc. |
| Text Generation | XSUM, Semantic Content, CCD, IWSLT14, WMT14, ROCStories, E2E, QQP, Wiki-Auto, Quasar-T, AG News Topic | Rouge, Semantic Acc, Mem, BLEU, Div, BERTScore, SacreBLEU, MAUVE Score, Content Fluency, POS | Chen et al. (2023a), Yuan et al. (2024), Chen et al. (2022a), Lovelace et al. (2024), etc. |
| 3D Generation | BraTS2020, ShapeNet, Objaverse, Cap3D, LLFF, HumanML3D, AMASS, KIT, HumanAct12, IBRNet, Instruction-NeRF2NeRF | Dice, HD95, CD, EMD, 1-NNA, COV, CLIP, Aesthetic, Similarity, R-Precision, FID, DIV, MM-Dist, ACC, Diversity, MModality | Bieder et al. (2023), Mo (2024), Li et al. (2024c), Park et al. (2023), Yu et al. (2024b), etc. |

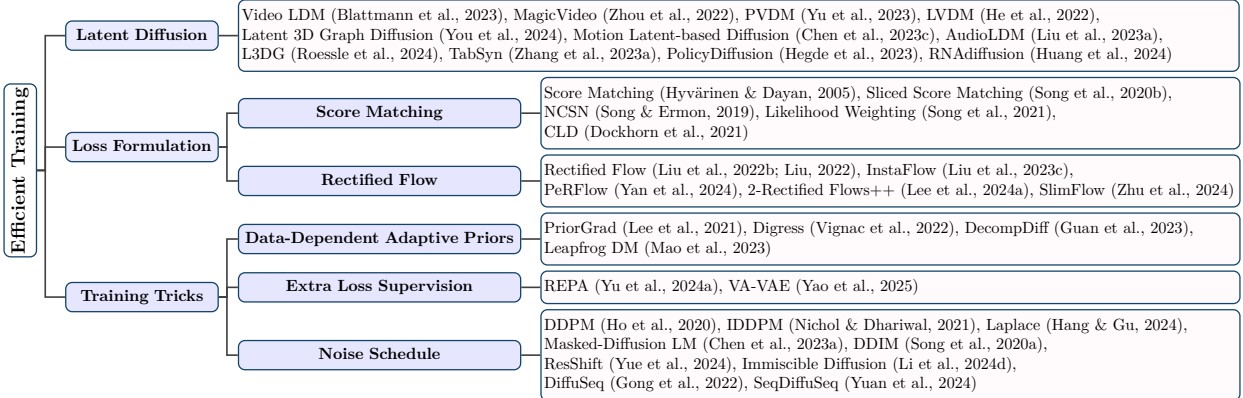

Figure 3: Summary of efficient training techniques for diffusion models.

## 3 Algorithm-Level Efficiency Optimization

### 3.1 Efficient Training

Efficient training aims to optimize the training process of DMs, reducing computational costs while accelerating convergence. As summarized in Figure 3, enhancing the efficiency of pre-training can be achieved through different and complementary techniques, including latent diffusion, loss formulation, and specialized training tricks. Latent diffusion models compress the optimization process by operating in lower-dimensional latent spaces, though they occasionally struggle with fine-grained detail reconstruction. Loss formulation methods enhance gradient estimation and stability, though sometimes facing numerical challenges. Meanwhile, various training tricks, including data-dependent adaptive priors, extra loss supervision, and optimized noise schedules, further enhance efficiency by leveraging problem-specific knowledge, though each introduces additional hyperparameters requiring careful tuning.

#### 3.1.1 Latent Space

Early diffusion models, such as DDPM (Ho et al., 2020), operate directly in pixel space, where images are generated through iterative noise addition and denoising. While these models achieve high-quality results, their reliance on pixel space introduces significant inefficiencies. Each diffusion step involves operations in a high-dimensional space, leading to substantial computational and memory overhead, especially for high-resolution images (e.g., 512×512). Additionally, the requirement for hundreds to thousands of iterative steps hinders real-time applications. Although methods like DDIM (Song et al., 2020a) have been proposed to speed up sampling, the pixel-level processing remains a fundamental bottleneck. In contrast, leveraging latent space significantly enhances training efficiency by operating in a lower-dimensional, compact representation, reducing both computational complexity and memory consumption while maintaining high-quality generation.

As research progresses, researchers have begun exploring compromise approaches to circumvent the high-dimensional pixel space. Consequently, various methods for introducing the latent space have increasingly gained attention from the academic community. To reduce computational complexity, researchers have introduced autoencoders (AEs) and their variants to compress images into a lower-dimensional latent space. Among them, Variational Autoencoders (VAEs ( Kingma et al. (2013))) map images to a Gaussian distribution in the latent space via an encoder, while a decoder reconstructs images from latent variables. Compared to traditional AEs, VAEs learn the data distribution rather than specific samples, mitigating overfitting. However, the generated images often appear blurry due to insufficient latent space constraints, as the KL regularization weight is relatively low. Vector Quantized Variational Autoencoders (VQ-VAEs ( Van Den Oord et al. (2017))) enhance compression efficiency by introducing a discrete latent space through vector quantization. The integration of compression techniques with diffusion models has given rise to various latent space-based diffusion models, with Latent Diffusion Models (LDMs) being the most prominent.

Benefiting from the efficiency advantages of the latent space and the cross-modal cross-attention mechanism of LDMs, a diverse range of image generation(eg. Stable Diffusion( Rombach et al. (2022))) and image editing models(eg. DiffEdit( Couairon et al. (2022))) has emerged.

Beyond traditional image synthesis and editing, various other diffusion model-based tasks achieve computational efficiency through the introduction of latent spaces, too. For videos with high-dimensionality, complex temporal dynamics and large spatial variations, learning video distributions within a low-dimensional latent space has proven to be an effective method for efficiently generating high-quality videos. For instance, Video LDM (Blattmann et al., 2023), MagicVideo (Zhou et al., 2022), PVDM (Yu et al., 2023) and LVDM (He et al., 2022) apply the latent diffusion model paradigm to video generation. Video LDM focuses on high-resolution video generation. Initially, the LDM is pretrained exclusively on images; subsequently, turn the image generator into a video generator by introducing a temporal dimension to the latent space diffusion model and fine-tuning on encoded image sequences, i.e., videos. LVDM proposes hierarchical diffusion in the latent space such that longer videos with more than one thousand frames can be produced. To further overcome the performance degradation issue for long video generation, LVDM introduces conditional latent perturbation and unconditional guidance. You et al. (You et al., 2024) perform 3D graph diffusion in a low-dimensional latent space, which is learned through cascaded 2D–3D graph autoencoders for low-error reconstruction and symmetry-group invariance, resulting in training that is an order of magnitude faster. Motion Latent-based Diffusion (Chen et al., 2023c) is able to produce vivid motion sequences conforming to the given conditional inputs and substantially reduce the computational overhead in both the training and inference stages by performing diffusion process on the motion latent space. AudioLDM (Liu et al., 2023a), a text-to-audio system built on a latent space that learns continuous audio representations from contrastive language-audio pretraining latents, offers advantages in both generation quality and computational efficiency. L3DG (Roessle et al., 2024) utilizes a compressed latent space learned by a vector-quantized variational autoencoder, coupled with a sparse convolutional architecture to efficiently operate on room-scale scenes. As a result, the complexity of the costly 3D Gaussians generation process through diffusion is significantly reduced. TabSyn (Zhang et al., 2023a) attempt to utilize a diffusion model with a carefully crafted latent space in tabular data synthesis tasks, discovering that generation quality significantly improves while synthesis speed also increases. In reinforcement learning, condensing the archive into a single model while retaining the performance and coverage of the original collection of policies has proved challenging. Hegde et al. (Hegde et al., 2023) propose using latent diffusion models to distill the archive into a single generative model over policy parameters, achieving a compression ratio of 13x. RNAdiffusion (Huang et al., 2024) compresses token-level, biologically meaningful representations of RNA sequences into a set of fixed-length latent vectors and reconstructs RNA sequences from these latent variables. It utilizes a latent diffusion model to achieve controllable and efficiently translated RNA sequence generation.

### 3.1.2 Loss Formulation

In this section, we examine methods that enhance the efficiency of different loss formulations in diffusion models. For score matching (Hyvärinen & Dayan, 2005), we present approaches to reduce its computational costs. For rectified flow (Liu et al., 2022b), we explore how its carefully designed formulation enables straight-line sampling trajectories and subsequent improvements that further increase efficiency. We also include Flow Matching's Optimal Transport approach (Lipman et al., 2022) in the rectified flow discussion, as it similarly achieves direct trajectory learning through linear parameterization of probability paths.

**Score Matching.** Compared to DDPM's straightforward optimization in Eq.(6), although score matching Eq.(9) avoids the computation of the partition function $Z_{\boldsymbol{\theta}}$, it still faces computational challenges, particularly in high-dimensional data. The computation of the trace of the Hessian matrix substantially increases the complexity as the dimensionality grows. Specifically, computing the trace requires many more backward passes than the gradient, making score matching computationally expensive for high-dimensional data.

Therefore, to address the computational inefficient issue of training process, Song et al. (2020b) observed that one-dimensional problems are typically much easier to solve than high-dimensional ones. Inspired by the idea of the Sliced Wasserstein Distance (Rabin et al., 2012), they proposed Sliced Score Matching. The core idea of sliced score matching is to project both the score function of the model $\boldsymbol{s}_m(\mathbf{x}; \boldsymbol{\theta})$ and the data $\boldsymbol{s}_d(\mathbf{x})$ onto a random direction $\mathbf{v}$, and compare the differences along that direction. The sliced score matching

objective is defined as:

$$L(\boldsymbol{\theta}; p_{\mathbf{v}}) = \frac{1}{2}\mathbb{E}_{p_{\mathbf{v}}}\mathbb{E}_{p_d(\mathbf{x})}\left[\left(\mathbf{v}^\top \boldsymbol{s}_m(\mathbf{x}; \boldsymbol{\theta}) - \mathbf{v}^\top \boldsymbol{s}_d(\mathbf{x})\right)^2\right] \tag{22}$$

To eliminate the dependence on $\boldsymbol{s}_d(\mathbf{x})$, integration is applied by parts, similar to traditional score matching, resulting in the following form:

$$J(\boldsymbol{\theta}; p_{\mathbf{v}}) = \mathbb{E}_{p_{\mathbf{v}}}\mathbb{E}_{p_d(\mathbf{x})}\left[\mathbf{v}^\top \nabla_\mathbf{x} \boldsymbol{s}_m(\mathbf{x}; \boldsymbol{\theta})\mathbf{v} + \frac{1}{2}(\mathbf{v}^\top \boldsymbol{s}_m(\mathbf{x}; \boldsymbol{\theta}))^2\right] \tag{23}$$

which achieves scalability by reducing the complexity of the problem by projecting high-dimensional score functions onto low-dimensional random directions, thereby avoiding the full Hessian computation.

While effective for dimensionality reduction, score estimation still faces challenges in low data density regions where data samples are sparse.

Building upon sliced score matching, to address the issue of inaccurate score estimation in low data density regions, Song & Ermon (2019) introduces a novel generative framework that employs Langevin dynamics to produce samples based on estimated gradients of the data distribution $p_{\text{data}}(\mathbf{x})$. They proposed Noise Conditional Score Networks (NCSN) $s_\theta(\mathbf{x}, \sigma)$, which jointly estimate scores across multiple noise-perturbed data distributions. By conditioning on a geometric sequence of noise levels $\sigma_3 > \sigma_2 > \sigma_1$, a single network learns to estimate scores for distributions ranging from highly smoothed $p_{\sigma_3}(\mathbf{x})$ that fill low-density regions to concentrated $p_{\sigma_1}(\mathbf{x})$ that preserve the structure of the original data manifold. This unified training approach enables robust score estimation across the entire data space. Following a similar derivation, as Song et al. (2021), Dockhorn et al. (2021) introduces Coupled Langevin Dynamics (CLD),

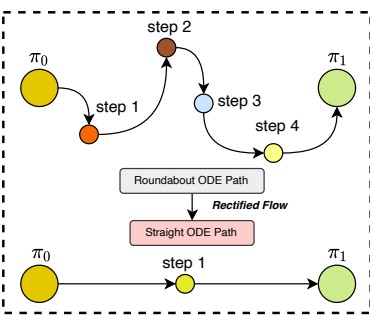

Figure 4: Illustration of the rectified flow.

redefining the score matching objective within the CLD framework. Unlike traditional score matching methods that inject noise directly into the data space, CLD simplifies the task by only requiring the model to learn the score of the conditional distribution $p_t(v_t \mid x_t)$, where noise is injected into an auxiliary variable $v_t$ coupled with the data.

**Rectified Flow.** As illustrated in Figure 4, Rectified Flow, proposed by (Liu et al., 2022b; Liu, 2022), introduces a method for training ordinary differential equation (ODE) models by learning straight transport paths between two distributions, $\pi_0$ and $\pi_1$.

The key idea is to minimize the transport cost by ensuring that the learned trajectory between these two distributions follows the most direct route, which can be computationally efficient to simulate. Unlike traditional diffusion models, which may follow roundabout paths, Rectified Flow leverages a simpler optimization problem to create a straight flow, thereby improving both training efficiency and the quality of the generated samples. Flow Matching (Lipman et al., 2022), as discussed in §2.1.4, introduces the Optimal Transport (OT) approach as a significant advancement in generative modeling. The key innovation of the OT method lies in how it parameterizes conditional probability paths. It represents conditional probability paths in Eq(19) using Gaussian distributions with means $\mu_t(x) = tx_1$ and standard deviations $\sigma_t(x) = 1 - (1 - \sigma_{\min})t$ that change linearly with time. This linear parameterization produces a vector field with constant direction, enabling transitions from noise to data along straight-line trajectories as well.

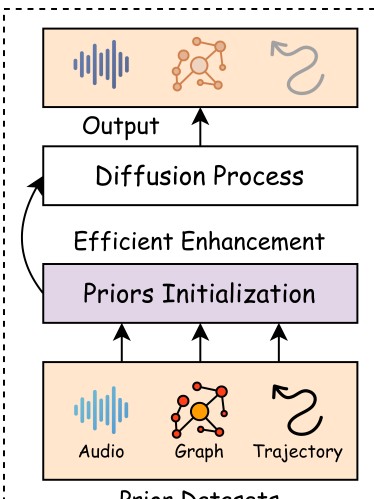

Figure 5: Illustration of data-dependent adaptive priors for diffusion processes across different modalities, demonstrating how tailored priors improve generation on quality.

Building upon the foundation of Rectified Flow, InstaFlow (Liu et al., 2023c) applies the Rectified Flow concept to text-to-image generation, achieving a significant breakthrough. InstaFlow represents a major advancement in efficient diffusion models, which are capable of high-quality image generation in just one step.

It applied Rectified Flow to large-scale datasets and complex models like Stable Diffusion, introduced a novel text-conditioned pipeline for one-step image generation, and achieved an FID score of 23.3 on MS COCO 2017-5k.

InstaFlow's success highlights the potential of Rectified Flow in dramatically reducing the computational cost of diffusion models while maintaining high output quality.

Following InstaFlow, Yan et al. (2024) proposed PeRFlow, further extending the Rectified Flow concept to create a more flexible and universally applicable acceleration method. PeRFlow divides the sampling process into multiple time windows, applying the reflow operation to each interval, creating piecewise linear flows that allow for more nuanced trajectory optimization. Through carefully designed parameterizations, PeRFlow models can inherit knowledge from pretrained diffusion models, achieving fast convergence and superior transfer ability. This approach positions PeRFlow as a universal plug-and-play accelerator compatible with various workflows based on pretrained diffusion models. While Rectified Flow showed great promise, there was still room for improvement, especially in low Number of Function Evaluations (NFE) settings. Addressing this, Lee et al. (2024a) focused on enhancing the training process of Rectified Flows. They discovered that a single iteration of the Reflow algorithm is often sufficient to learn nearly straight trajectories and introduced a U-shaped timestep distribution and LPIPS-Huber premetric to improve one-round training. These improvements led to significant enhancements in FID scores, particularly in low NFE settings, outperforming state-of-the-art distillation methods on various datasets. Most recently, Zhu et al. (2024) proposed SlimFlow, a method designed to address the joint compression of inference steps and model size within the Rectified Flow framework, introducing Annealing Reflow to address initialization mismatches between large teacher models and small student models, and developing Flow-Guided Distillation to improve performance on smaller student models.

### 3.1.3 Training Tricks

Training tricks encompass a range of practical strategies aimed at improving the efficiency, convergence, and sample quality of diffusion models by optimizing various aspects of the learning process. In this section, we explore three key approaches: data-dependent adaptive priors, which tailor initial distributions to specific data characteristics for faster convergence; extra loss supervision, which introduces additional loss terms to better align the model with data distributions and enhance generation quality; and noise schedule design, which governs the addition and removal of noise to streamline the diffusion process and accelerate sampling.

**Data-Dependent Adaptive Priors.** To enhance the training efficiency of diffusion models and improve the quality of generated samples, data-dependent adaptive priors can be utilized to tailor the prior distribution to specific tasks and datasets. This approach leverages priors that better align with the data distribution, thereby accelerating the training process and ensuring that generated samples more closely match the true data distribution. Recent studies have explored how data-dependent adaptive priors can improve the training of diffusion models.

As a method under efficient training (see Figure 5), data-dependent adaptive priors can be applied across various modalities, such as speech, graphs, and trajectories. By aligning the prior with the data distribution specific to each modality, the model can achieve faster convergence during training while producing outputs that better reflect the underlying data structure. In traditional diffusion models, the prior is typically assumed to be a standard Gaussian distribution $p(z) = \mathcal{N}(0, I)$. However, this assumption may not align well with the actual data distribution, potentially leading to inefficiencies in training. By constructing data-dependent adaptive priors based on the data $X$, the model can achieve better initialization during training, accelerating convergence without relying solely on the standard Gaussian assumption.

Lee et al. (2021) introduced PriorGrad, which enhances the training of diffusion models for speech synthesis by using an adaptive prior derived from conditional data statistics. This method significantly improves the efficiency of the denoising process during training, leading to faster convergence while enhancing the perceptual quality and robustness of generated samples, even with smaller network capacities.

Vignac et al. (2022) proposed DiGress, a discrete denoising diffusion model for graph generation. By leveraging data-dependent priors, this model better captures the discrete nature of graph data, improving training

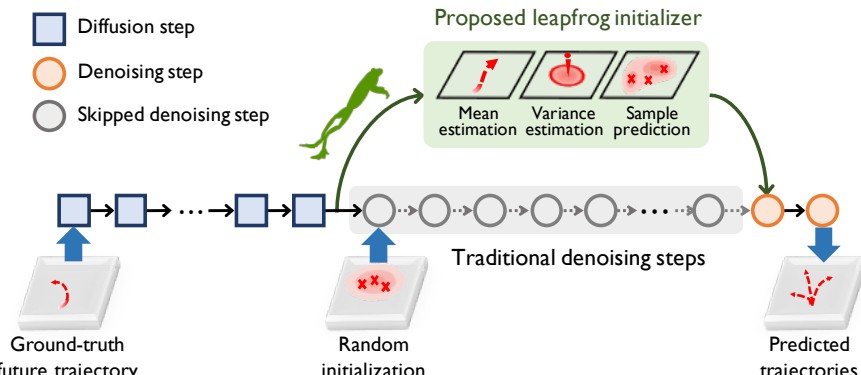

Figure 6: The Leapfrog diffusion model (Mao et al., 2023) accelerates inference by using a leapfrog initializer to approximate the denoised distribution, replacing extended denoising sequences while preserving representation capacity.

efficiency and the quality of generated graphs, particularly for applications like chemical molecular structures and social networks.

In drug design, Guan et al. (2023) introduced DecompDiff, which improves the training of diffusion models by using decomposed priors to separately model different structural components of drug molecules. This approach enhances the model's ability to capture molecular structure information during training, leading to the generation of more accurate drug candidates.

As shown in Figure 6, Mao et al. (2023) proposed the Leapfrog Diffusion Model for stochastic trajectory prediction, introducing a leapfrog initializer based on adaptive priors to skip multiple denoising steps, accelerating training while maintaining accuracy for real-time applications like autonomous driving. Building on this, Fu et al. (2025) developed MoFlow, a one-step flow matching model with IMLE-based distillation for human trajectory forecasting. MoFlow employs a novel flow matching loss to ensure accuracy and diversity in predicted trajectories, using data-dependent adaptive priors based on past trajectories and interactions to enhance alignment with the data distribution. Its IMLE distillation achieves a 100x faster one-step student model with comparable performance. Similarly, Jiang et al. (2024) introduced SceneDiffuser, a scene-level diffusion model using amortized diffusion to optimize efficiency for driving simulation, supporting both initialization and rollout. These advances highlight the role of adaptive priors in boosting training efficiency for real-time multimodal applications.

**Extra Loss Supervision.** To further enhance the training efficiency and sample quality of diffusion models, incorporating additional loss supervision has proven to be an effective strategy. Beyond standard denoising objectives, extra loss terms can guide the model toward better alignment with data distributions, accelerate convergence, and improve the robustness of generated outputs. Recent advancements have demonstrated the power of tailoring loss functions to leverage external knowledge or address specific optimization challenges.

One notable approach introduces REPresentation Alignment (REPA) (Yu et al., 2024a), a regularization technique that aligns the hidden states of the denoising network with representations from a pre-trained visual encoder, such as DINOv2. By adding a loss term that enforces this alignment, REPA ensures that the diffusion Transformer (e.g., DiT or SiT) leverages high-quality external visual priors rather than learning representations from scratch. This supervision accelerates training—achieving over $17.5\times$ speedup in some cases—and boosts generation quality, reaching a state-of-the-art FID of 1.42 on ImageNet 256x256 with classifier-free guidance. The additional loss acts as a bridge between the model's internal learning and established visual knowledge, making training both faster and more effective.

Similarly, another method tackles the optimization dilemma in latent diffusion models (LDMs) by introducing extra loss supervision through the VA-VAE framework (Yao et al., 2025). This approach aligns the latent space of a variational autoencoder with a pre-trained vision foundation model, using a loss term to ensure the latent representations capture the data distribution more effectively. This supervision mitigates the trade-off between reconstruction fidelity and generation quality, a common challenge in LDMs where increasing latent dimensionality often hampers generative performance. Combined with an optimized LightningDiT architecture, this method achieves an FID of 1.35 on ImageNet 256x256, with training convergence accelerated by

over 21× compared to baseline DiT models. The extra loss supervision in VA-VAE enhances the model's ability to balance reconstruction and generation tasks, leading to both efficiency gains and superior sample quality.

**Noise Schedule.** Noise schedule is a crucial component of diffusion models, governing how noise is added during the forward process and removed during the reverse process. Optimizing the noise schedule can significantly enhance the training efficiency of diffusion models by enabling faster convergence and more effective learning of data distributions. Denoising Diffusion Probabilistic Models (DDPM) (Ho et al., 2020) introduced a linear noise schedule that gradually decreases the variance of the noise added in the forward process, as defined in Eq.(1). However, the linear schedule requires calculating complex noise terms across numerous timesteps, which can slow down the training process and highlight the need for more efficient noise schedule designs. As shown in Figure 7, efficient noise schedules can be classified into two main categories: systematic noise addition and dynamic noise adjustment.

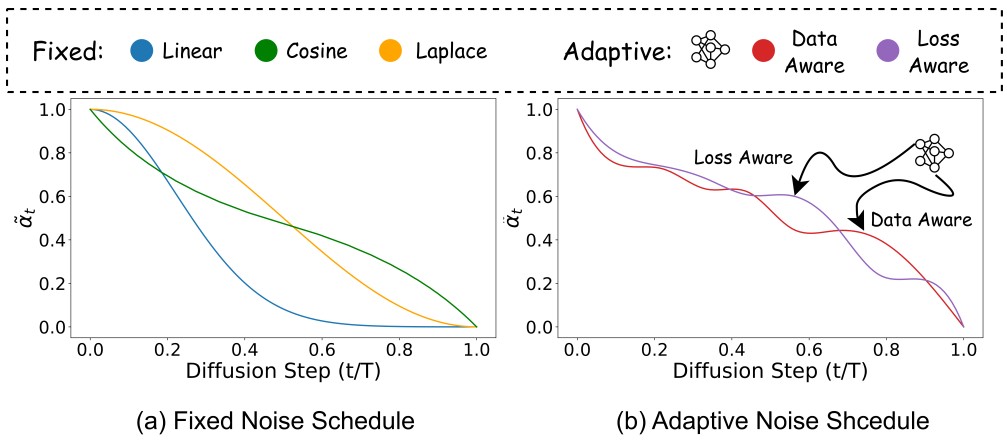

(a) Fixed Noise Schedule      (b) Adaptive Noise Shcedule

Figure 7: Illustration of two categories of noise schedules.

One approach involves systematically adding noise during the training process at predefined intervals or according to specific levels. DDPM (Ho et al., 2020) employs a linear noise schedule, where the noise variance changes deterministically over time, serving as a foundational example. Building on this, the Improved Denoising Diffusion Probabilistic Model (IDDPM) (Nichol & Dhariwal, 2021) introduces a cosine noise schedule, defined as

$$\beta_t = 1 - \frac{\cos\left(\frac{t/T+s}{1+s} \cdot \frac{\pi}{2}\right)}{\cos\left(\frac{s}{1+s} \cdot \frac{\pi}{2}\right)}, \tag{24}$$

where $t$ is the current timestep, $T$ is the total number of timesteps, and $s$ is a small positive constant for smoothing initial noise addition. The cosine noise schedule optimizes the noise distribution during training, enabling the model to learn data structures more effectively and achieve faster convergence. However, it allocates computational resources evenly across all noise intensities, which may not prioritize the most critical regions for training.

To address this inefficiency, Hang & Gu (2024) proposed the Laplace noise schedule, which enhances training efficiency by increasing the sampling frequency around critical noise regions. This approach ensures that the model focuses computational resources on medium noise intensities, which are more effective for learning data structures and removing noise, leading to faster convergence and improved sample quality during training. The Laplace schedule balances noise addition across timesteps, resulting in a more robust training process.

For text generation, traditional diffusion models often add uniform Gaussian noise to each word, which fails to leverage linguistic features and increases computational burden during training. To address this, Chen et al. (2023a) introduced a soft-masking noise strategy that gradually adds noise based on word importance, measured by term frequency and information entropy. Using a square-root noise schedule (Li et al., 2022), this method incrementally increases noise levels, stabilizing the training process and enabling the model to learn linguistic structures more effectively from the initial latent variable $X_0$ to noisy variables $X_{1:T}$.

In contrast, another set of methods dynamically adjusts the noise schedule based on the model's state or data during training. Song et al. (2020a) developed Denoising Diffusion Implicit Models (DDIM), which improve the noise schedule in DDPM by introducing a non-Markovian forward process, defined as

$$x_{t-1} = \sqrt{\alpha_{t-1}} \left( x_t - \sqrt{1 - \alpha_t} \epsilon_\theta(x_t) \right) + \sqrt{1 - \alpha_{t-1} - \sigma_t^2} \epsilon_\theta(x_t) + \sigma_t \epsilon_t, \tag{25}$$

where $\alpha_t$ controls the noise addition over time, and $\sigma_t$ is dynamically adjusted based on the current state $x_t$ and initial state $x_0$. This dynamic adjustment leverages the entire trajectory, enabling more precise noise control during training, which enhances the model's ability to learn data distributions efficiently and improves training convergence.

Inspired by DDIM, Yue et al. (2024) proposed ResShift, which introduces a noise schedule that constructs a shorter Markov chain by shifting residuals between high-resolution (HR) and low-resolution (LR) images instead of adding Gaussian noise. The noise schedule is defined as

$$\sqrt{\eta_t} = \sqrt{\eta_1} \times b_0^{\beta_t}, \quad t = 2, \ldots, T - 1, \tag{26}$$

where

$$\beta_t = \left( \frac{t-1}{T-1} \right)^p \times (T-1), \quad b_0 = \exp\left( \frac{1}{2(T-1)} \log \frac{\eta_T}{\eta_1} \right), \tag{27}$$

with $T$ as the total timesteps, $t$ as the current timestep, $p$ controlling the growth rate of $\sqrt{\eta_t}$, and $\eta_1$ and $\eta_T$ as the initial and final noise levels. This non-uniform noise progression allows the model to focus training on key regions, improving convergence and learning efficiency for super-resolution tasks.

To further optimize training, Li et al. (2024d) proposed Immiscible Diffusion, inspired by the physical phenomenon of immiscibility. Unlike traditional methods that diffuse each image across the entire noise space, this approach reassigns noise to images within a mini-batch to minimize the distance between image-noise pairs. By matching each image with nearby noise, Immiscible Diffusion reduces the complexity of denoising during training, enabling the model to learn data structures more efficiently.

For text generation, traditional methods (Gong et al., 2022) often use fixed noise schedules, requiring recalculation of the input sequence at each timestep, which leads to inefficient training. In contrast, Yuan et al. (2024) introduced a dynamic noise adjustment technique that records loss values at each timestep during training and uses linear interpolation to map these losses to noise schedule parameters. This adaptive approach optimizes noise levels at each timestep and token position, improving training efficiency and enabling the model to better capture linguistic features.

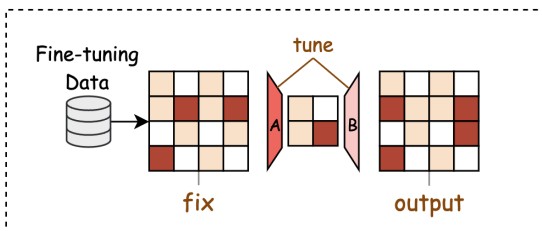

Figure 8: Illustration of Hu et al. (2021)'s reparameterization approach, where only parameters A and B are trained.

## 3.2 Efficient Fine-Tuning

Fine-tuning pre-trained diffusion models demands efficient resource use, leading to distinct strategies: LoRA, Adapters, and ControlNet. LoRA stands out for its low-rank parameter updates, slashing memory needs by up to 90% and enabling rapid inference, though it struggles with nuanced spatial control. Adapters, by contrast, inject lightweight modules for task-specific tweaks, excelling in conditional guidance (e.g., sketches via T2I-Adapter) but relying on input quality, which can limit robustness. ControlNet, however, leverages additional network branches for precise spatial conditioning, offering unmatched control over structure and style, yet at the cost of higher computational load—mitigated in variants like ControlNet-XS. These trade-offs are explored in detail in the subsections.

### 3.2.1 LoRA

Low-Rank Adaptation (LoRA) (Hu et al., 2021) is a model adaptation method that maintains frozen pre-trained model weights while enabling efficient task adaptation through the injection of low-rank decomposition matrices into each Transformer layer. The core mathematical foundation of this approach lies in

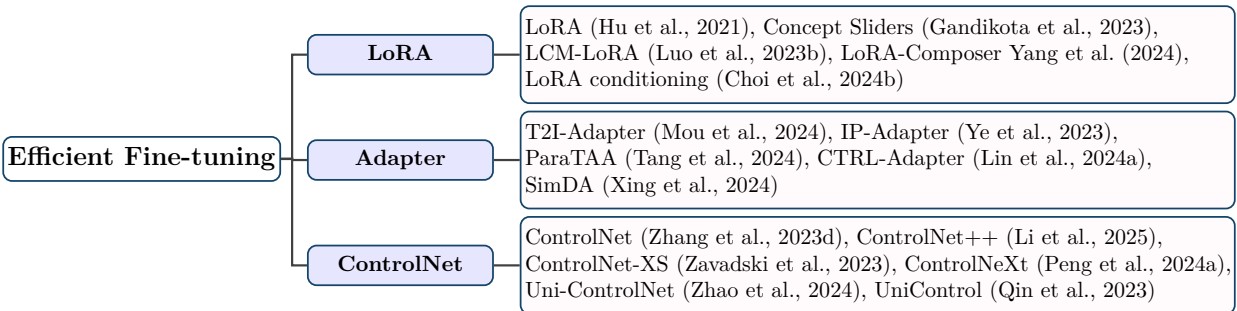

Figure 9: Summary of efficient fine-tuning techniques for diffusion models.

its representation of the weight update mechanism: for a pre-trained weight matrix $W_0 \in \mathbb{R}^{d \times k}$, LoRA represents the weight update as:

$$W = W_0 + \Delta W, \text{ where } \Delta W = BA \tag{28}$$

where $B \in \mathbb{R}^{d \times r}$ and $A \in \mathbb{R}^{r \times k}$ are trainable low-rank matrices, and the rank $r \ll \min(d, k)$. During forward propagation, for an input $x \in \mathbb{R}^k$, the model computes the hidden representation $h \in \mathbb{R}^d$ as:

$$h = W_0 x + \Delta W x = W_0 x + BA x \tag{29}$$

The complete process is illustrated in Figure 8. A key advantage of this design lies in its deployment efficiency, where the explicit computation and storage of $W = W_0 + BA$ enables standard inference procedures without introducing additional latency. Originally proposed for fine-tuning Large Language Models (LLMs), LoRA has demonstrated remarkable parameter efficiency and memory reduction in model adaptation. While predominantly utilized in LLM fine-tuning, recent research has extended its application to diffusion models, indicating its potential as a versatile adaptation technique across different deep learning architectures. LCM-LoRA (Luo et al., 2023b) proposes a universal acceleration approach for diffusion models. As shown in Figure 10, this method achieves fast sampling by adding an Acceleration vector $\tau_{LCM}$ to the Base LDM Rombach et al. (2022). This module adopts LoRA (Hu et al., 2021) to attach low-rank matrices to the original model without architectural modifications. For customized diffusion models that are fine-tuned for specific text-to-image generation tasks, the task-specific LoRA ($\tau'$) and acceleration LoRA ($\tau_{LCM}$) can be linearly combined through Eq.(30) to achieve fast inference while maintaining generation quality. More importantly, it provides a plug-and-play solution that reduces sampling steps from dozens to around 4, while maintaining compatibility with any pre-trained text-to-image diffusion model.

$$\tau'_{LCM} = \lambda_1 \tau' + \lambda_2 \tau_{LCM} \tag{30}$$

Beyond the acceleration achieved by LCM-LoRA, Concept Sliders (Gandikota et al., 2023) extends LoRA's application to precise control over image generation attributes. This method identifies low-rank directions in the diffusion parameter space corresponding to specific concepts through LoRA adaptation. The method freezes the original model parameters and trains a LoRA adapter to learn concept editing directions. Given an input $(x_t, c_t, t)$, where $x_t$ is the noisy image at timestep $t$. For a target concept $c_t$, the model is guided through a score function to enhance certain attributes $c_+$ while suppressing others $c_-$. This training objective can be formulated as:

$$\epsilon_{\theta*}(x, c_t, t) \leftarrow \epsilon_\theta(x, c_t, t) + \eta(\epsilon_\theta(x, c_+, t) - \epsilon_\theta(x, c_-, t)) \tag{31}$$

Here, $\epsilon_\theta$ represents the denoising model's prediction, and $\eta$ is a guidance scale. With this formulation, the method enables smooth control over concept strength through the guidance scale $\eta$ while maintaining concept independence in the learned directions. By leveraging LoRA's parameter-efficient nature, it achieves precise attribute manipulation with minimal computational overhead.

Besides, LoRA-Composer Yang et al. (2024) advances LoRA's application in diffusion models toward seamless multi-concept integration. While previous works focus on acceleration or single-concept control, this

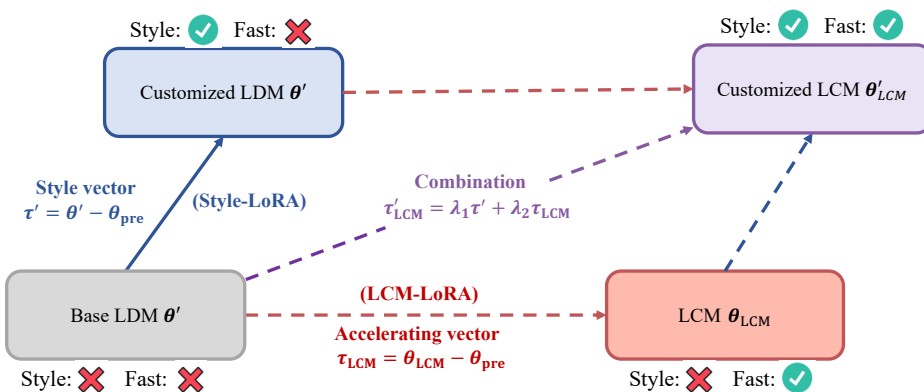

Figure 10: Illustration of LCM-LoRA (Luo et al., 2023b).

approach tackles the challenging task of combining multiple LoRA-customized concepts within a single image generation process. It combines multiple LoRAs in diffusion models by modifying the U-Net's attention blocks. Specifically, it enhances both cross-attention and self-attention layers within U-Net to enable direct fusion of multiple LoRAs. Compared to traditional methods like Mix-of-Show Gu et al. (2024) that require training a fusion matrix to merge multiple LoRAs, which increases computational overhead and may degrade generation quality. It directly combines multiple lightweight LoRAs through modified attention blocks, avoiding the overhead of retraining models. While LoRA-Composer focuses on fusing multiple LoRAs for multi-concept control, Choi et al. (2024b) explores the fundamental application of LoRA in attention layers. Both these works enhance diffusion models by modifying the attention mechanism in U-Net. The latter proposes a structured conditioning approach in U-Net blocks with three key components: (1) conventional convolutional blocks using scale-and-shift conditioning for feature normalization adjustment, (2) attention blocks enhanced by LoRA adapters that condition both QKV computation and projection layers through learnable low-rank matrices, and (3) two LoRA conditioning implementations - TimeLoRA/ClassLoRA for discrete-time settings and UC-LoRA for continuous SNR settings, which utilize MLP-generated weights to combine multiple LoRA bases. Them method achieves improved performance over traditional conditioning while only increasing the parameter count by approximately 10% through efficient low-rank adaptations in the attention layers.

### 3.2.2 Adapter

Adapters are lightweight modules designed to enable efficient task adaptation by introducing small network layers into pre-trained models, allowing task-specific feature learning while keeping the original weights frozen. As illustrated in Figure 11, adapter layers are placed within the transformer block, positioned between normalization and feed-forward layers. Each adapter module consists of a down-projection, nonlinearity, and up-projection, which generates task-specific transformations without altering the core model's structure.

This design significantly reduces memory and computational requirements, making adapters well-suited for tasks requiring lightweight parameter updates, such as text-to-image generation (T2I) and simulated domain adaptation (SimDA).

T2I-Adapter (Mou et al., 2024) is an adapter designed to enhance control in text-to-image generation models by introducing conditional features such as sketches, depth maps, and semantic segmentation maps, allowing for structural guidance in generated images. Unlike approaches that require modifying the model's core architecture, T2I-Adapter uses lightweight modules to incorporate external condition information into the generation process without altering the pre-trained model itself. This method improves the accuracy and consistency of generated images without increasing computational costs. In implementation, T2I-Adapter employs convolutional and residual blocks to align conditional inputs with the spatial dimensions of intermediate features in the UNet model, thus capturing structural information at multiple scales. This integration allows T2I-Adapter to flexibly incorporate conditional features, such as sketches and depth maps, providing

enhanced control over text-to-image generation. Such multi-adapter strategies are particularly suitable for tasks requiring high customization in image generation, enabling simultaneous input of various structural features to refine the output.

IP-Adapter (Ye et al., 2023) enhances the consistency and visual quality of text-to-image generation by incorporating image prompts. Unlike T2I-Adapter (Mou et al., 2024), which provides structural guidance through sketches or depth maps, IP-Adapter focuses on capturing visual details from an input image, making it ideal for tasks requiring high visual consistency with a reference image. This adapter processes the input image prompt into latent features, allowing the generation model to capture visual information from the target image and maintain detail alignment throughout the generation process. In its workflow, the image prompt is first mapped into the latent space and then processed through convolution and normalization modules within the adapter, enabling the model to utilize these features during inference. This setup enables the generation model to draw rich visual information from the image prompt, making IP-Adapter particularly suitable for tasks requiring high detail consistency, such as generating images with a style similar to the input image.

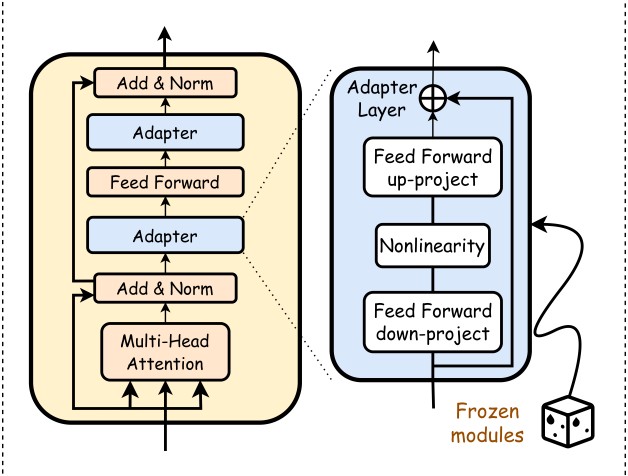

Figure 11: Architecture of the Adapter module: demonstrating the integration of adapter layers within a transformer block to achieve efficient task adaptation by adding lightweight transformations, while keeping the core model weights frozen.

CTRL-Adapter (Lin et al., 2024a) is designed to enhance attribute control during generation by guiding specific attributes such as emotion or object type, enabling precise customization in generated results. Unlike T2I-Adapter (Mou et al., 2024) and IP-Adapter (Ye et al., 2023), which focus on structural and detail consistency respectively, CTRL-Adapter is tailored to provide diversity control for the generation model. For example, as illustrated in Figure 12, the IP-Adapter architecture employs a decoupled

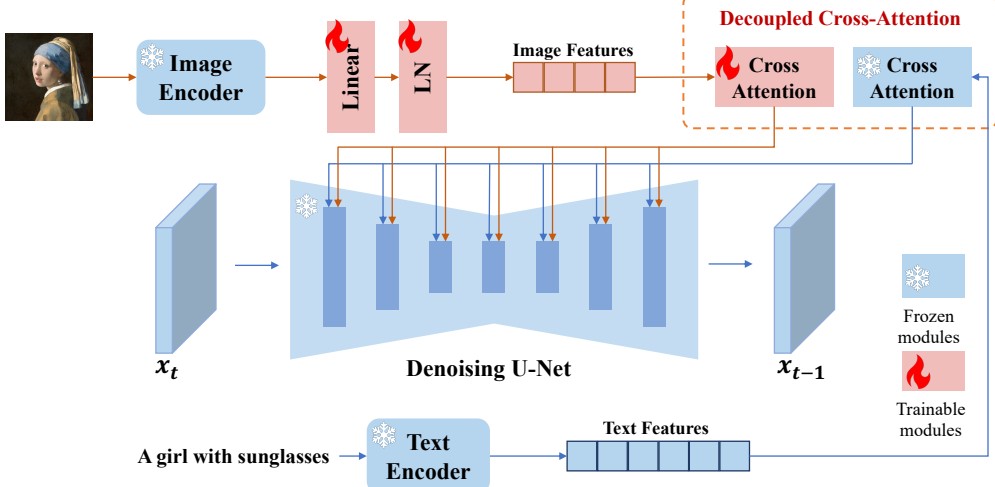

Figure 12: Architecture of IP-Adapter (Ye et al., 2023) using a decoupled cross-attention strategy, where only newly added modules are trained, and the pre-trained text-to-image model remains frozen.

cross-attention strategy, where only newly added modules are trained while the pre-trained text-to-image model remains frozen. In contrast, CTRL-Adapter can adjust the style of generated images based on specified emotions or object types, achieving controllable content generation without altering the core architecture of the model. This makes CTRL-Adapter particularly suitable for tasks requiring high customization in generation, such as emotion-driven text generation or stylized image synthesis.

SimDA (Xing et al., 2024) is an adapter suited for cross-domain generation tasks, achieving domain adaptation by utilizing simulated data within the adapter to enhance the model's performance on previously unseen data distributions. Unlike CTRL-Adapter (Lin et al., 2024a), which primarily focuses on attribute control, SimDA is designed to improve the model's generalization ability, allowing it to generate high-quality content even in unfamiliar data environments. SimDA is particularly useful in generation tasks that require domain transfer, such as adapting a model trained on one image dataset to perform well on another dataset. This enables the model to align with new data characteristics without compromising generation quality.

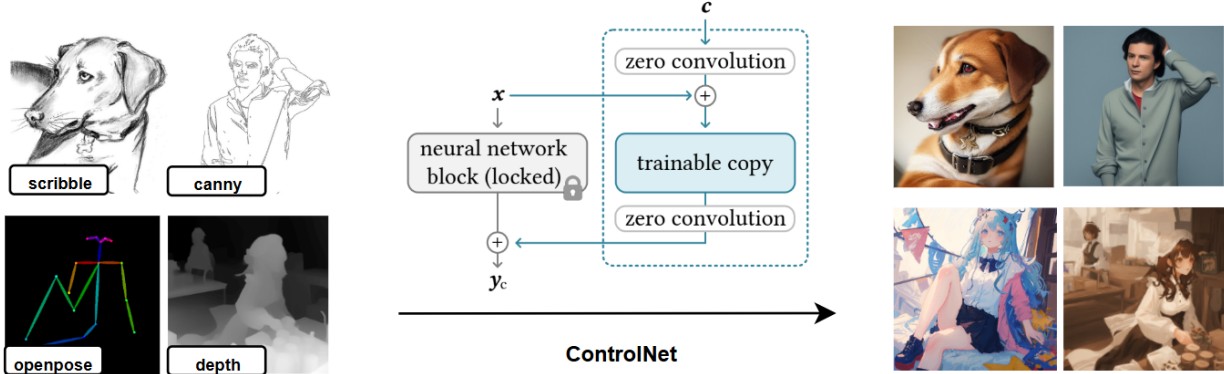

Figure 13: Illustration of ControlNet.

### 3.2.3 ControlNet

ControlNet (Zhang et al., 2023d) and its derivatives represent a significant advancement in adding spatial conditioning controls to pre-trained text-to-image diffusion models. The original ControlNet architecture (Zhang et al., 2023d), as illustrated in Figure 13, presents a novel approach to integrating various spatial conditions—such as scribbles, edge maps, open-pose skeletons, or depth maps—into the generative process while preserving the robust features of pre-trained diffusion models. The architecture employs zero convolution layers that gradually develop parameters without disrupting the pre-trained model's stability. This design enables versatile conditioning, allowing the model to effectively leverage different types of spatial information. Through these conditioning methods, ControlNet demonstrates a remarkable ability to guide generation with fine-grained control over structure, style, and composition. Building upon this foundation, several works have proposed improvements and alternatives. ControlNet++ (Li et al., 2025) addresses the challenge of alignment between generated images and conditional controls by introducing pixel-level cycle consistency optimization. Through a pre-trained discriminative reward model and an efficient reward strategy involving single-step denoised images, it achieves significant improvements in control accuracy, with notable gains in metrics such as mIoU (11.1%), SSIM (13.4%), and RMSE (7.6%) across various conditioning types. ControlNet-XS (Zavadski et al., 2023) reimagines the control system by enhancing the communication bandwidth between the controlling network and the generation process. This redesign not only improves image quality and control fidelity but also significantly reduces the parameter count, resulting in approximately twice the speed during both inference and training while maintaining competitive performance in pixel-level guidance tasks. The field has also seen efforts to unify multiple control capabilities. UniControl (Qin et al., 2023) introduces a task-aware HyperNet approach that enables a single model to handle diverse visual conditions simultaneously. Similarly, Uni-ControlNet (Zhao et al., 2024) proposes a unified framework supporting both local controls and global controls through just two additional adapters, significantly reducing training costs and model size while maintaining high performance. Most recently, ControlNeXt (Peng et al., 2024a) has pushed the boundaries of efficiency even further by introducing a streamlined architecture that minimizes computational overhead. It replaces the traditional heavy additional branches with a more concise structure and introduces Cross Normalization (CN) as an alternative to zero convolutions. This approach achieves fast and stable training convergence while reducing learnable parameters by up to 90% compared to previous methods.

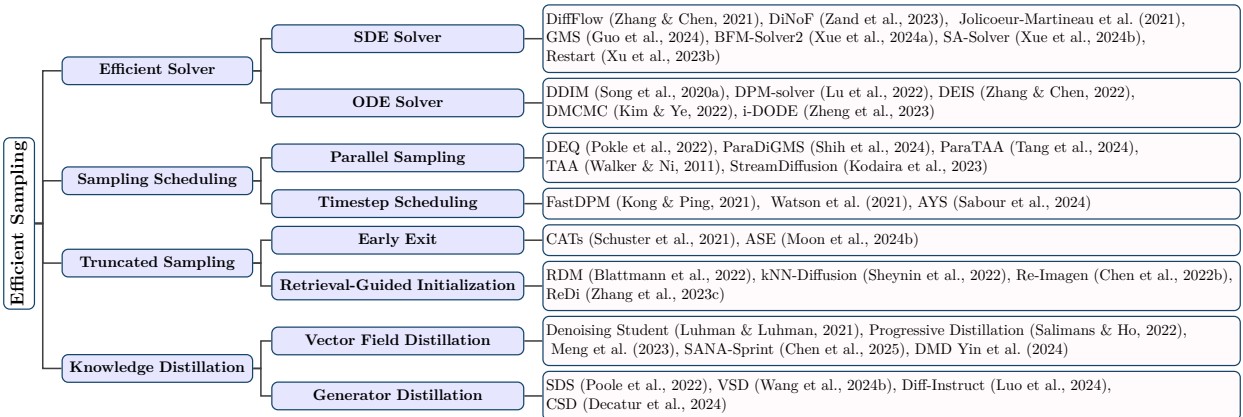

Figure 14: Summary of efficient sampling techniques for diffusion models.

## 3.3 Efficient Sampling

The standard diffusion sampling process is computationally intensive because it requires sequentially executing a large number of denoising steps, with each step dependent on the output of the previous one as formulated in Eq(4). This inherent sequential dependency makes the sampling procedure time-consuming, resulting in significantly slower generation. To address these computational challenges, researchers have developed efficient sampling methods through four principal approaches. As illustrated in Figure 14, these encompass efficient SDE and ODE solvers that reduce the required number of function evaluations, advanced sampling scheduling strategies including parallel sampling techniques and timestep optimization methods, knowledge distillation techniques that transfer diffusion model capabilities to more efficient representations, and truncated sampling approaches that leverage early exit mechanisms and retrieval-based techniques to further accelerate the generation process while preserving output quality.

### 3.3.1 Efficient Solver

As we mentioned in Section. §2.1.3, although diffusion solvers avoid the need to compute the intractable partition function, they still require numerous function evaluations with fixed step sizes for SDEs in Eq(11) or generic numerical integration methods for ODEs in Eq.(12), resulting in slow sampling procedures and substantial computational costs. These limitations create substantial computational costs and lengthy generation times. In the following sections, we present advances in both SDE and ODE methods that address these efficiency constraints.

**SDE Solver.** Nowadays, there are many ways to efficiently implement SDE-based solvers. (Zhang & Chen, 2021)introduces a novel generative modeling and density estimation algorithm called Diffusion Normalizing Flow (DiffFlow). Similar to the SDE of diffusion models Eq.(10), the DiffFlow model also has a forward process:

$$d\mathbf{x} = \mathbf{f}(\mathbf{x}, t, \theta)dt + g(t)d\bar{\mathbf{w}} \tag{32}$$

and a backward process:

$$d\mathbf{x} = \left[\mathbf{f}(\mathbf{x}, t, \theta) - g^2(t)\mathbf{s}(\mathbf{x}, t, \theta)\right]dt + g(t)d\bar{\mathbf{w}} \tag{33}$$

As a result of the learnable parameter $\theta$, the drift term $f$ is also learnable in DiffFlow, compared to the fixed liner function as in most diffusion models. Besides, these SDEs are jointly trained by minimizing the KL divergence. This allows the model to better adapt to changes in the data distribution, thus speeding up the convergence of the backward diffusion process. Similar to DiffFlow, Zand et al. (2023) proposes a method called Diffusion with Normalizing Flow priors that also combines diffusion models with normalizing flows. The method first uses a linear SDE in the forward process to convert the data distribution into a noise distribution gradually. In the reverse process, a normalizing flow network is introduced to map the standard Gaussian distribution to latent variables close to the data distribution through a series of reversible

transformations, which allows the samples to return to the data distribution more quickly, rather than relying on a large number of small incremental adjustments.

However, the fixed step sizes in existing SDE solvers Eq.(10), which usually require tremendous iterative steps, significantly affect generation efficiency. To address this, Jolicoeur-Martineau et al. (2021) proposes a novel adaptive step-size SDE solver that dynamically adjusts the step size based on error tolerance, thereby reducing the number of evaluations. Specifically, the proposed method dynamically adjusts the step size by estimating the error between first-order and second-order approximations, leveraging a tolerance mechanism that incorporates both absolute and relative error thresholds. Furthermore, the use of extrapolation enhances precision without incurring additional computational overhead. This approach obviates the need for manual step-size tuning and is applicable across a range of diffusion processes, including Variance Exploding and Variance Preserving models. As a result of Gaussian assumption for reverse transition kernels becomes invalid when using limited sampling steps. The Gaussian Mixture Solver (GMS) (Guo et al., 2024) optimized SDE solver by using Gaussian mixture distribution. It addresses the limitations of the traditional process of SDE solvers in Eq.(11), which assume a Gaussian distribution for the reverse transition kernel. Specifically, GMS replaces the Gaussian assumption with a more flexible Gaussian mixture mode and utilizes a noise prediction network with multiple heads to estimate the higher-order moments of the reverse transition kernel. At each sampling step, it employs the Generalized Method of Moments to optimize the parameters of the Gaussian mixture transition kernel, allowing for a more accurate approximation of the true reverse process, even with a limited number of discretization steps.

Instead, Xue et al. (2024a) unifies Bayesian Flow Networks (BFNs) with Diffusion Models (DMs) by introducing time-dependent SDEs into the BFN framework. BFNs work by iteratively refining the parameters of distributions at different noise levels through Bayesian inference, rather than directly refining the samples as in traditional diffusion models. To achieve theoretical unification between BFNs and DMs, the authors introduce a time-dependent linear SDE that governs the noise addition process in BFNs. This forward process includes two time-dependent functions: one controlling the drift of parameters and another controlling their diffusion. Based on this forward equation, they derive a corresponding reverse-time SDE for generating data from noise. This reverse process combines the drift term with a score-based correction term. This reverse-time SDE directly aligns with the denoising process in diffusion models, enabling the BFN sampling process to effectively replicate the behavior of diffusion models.

By optimizing the solving process of SDE in Eq.(10), Stochastic Adams Solver (SA-Solver) (Xue et al., 2024b) was presented. It is an innovative method designed to efficiently sample from Diffusion SDEs in Diffusion Probabilistic Models (DPMs) (Ho et al., 2020). By addressing the significant computational burden of traditional samplers, SA-Solver achieves this through a clever combination of variance-controlled diffusion SDEs and a stochastic Adams method (Buckwar & Winkler, 2006), which is a multi-step numerical technique that leverages prior evaluations to enhance efficiency. The method introduces a noise control function $\tau(t)$, enabling dynamic adjustment of the noise injected during sampling, which in turn strikes an optimal balance between sampling speed and the quality of the generated data. Operating within a predictor-corrector framework, SA-Solver first makes an initial estimate through the predictor step and then refines this estimate using the corrector step, ensuring greater accuracy in the final output. This strategic integration significantly reduces the number of function evaluations required. Xu et al. (2023b) introduces Restart, a novel sampling algorithm for diffusion models that strategically combines the strengths of SDE and ODE solvers. By theoretically analyzing Wasserstein bounds, the authors demonstrate that SDEs contract accumulated errors via stochasticity, while ODEs excel in low-NFE regimes due to smaller discretization errors. Restart bridges these regimes by alternating between noise injection and deterministic ODE backward steps, decoupling stochasticity from drift updates to amplify error contraction while retaining ODE efficiency.

**ODE Solver.** For efficiently implement the ODE-based solvers, Denoising Diffusion Implicit Models (DDIM) (Song et al., 2020a) builds upon the framework of Denoising Diffusion Probabilistic Models (DDPM) (Ho et al., 2020), offering significant enhancements in sampling efficiency, which is one of the first models to leverage ODEs explicitly for the accelerating sampling process.

$$q_\sigma(\mathbf{x}_{1:T}|\mathbf{x}_0) = q_\sigma(\mathbf{x}_T|\mathbf{x}_0) \prod_{t=2}^{T} q_\sigma(\mathbf{x}_{t-1}|\mathbf{x}_t, \mathbf{x}_0) \tag{34}$$

Unlike DDPM's Markovian forward process Eq.(1) where each state only depends on its immediate predecessor, DDIM utilizes the Non-Markovian Forward Process Eq.(34). These formulas allow each state not only to depend on its immediate predecessor but also on the initial state or a series of previous states. Specifically, Eq.(35) outlines how DDIM generates $\mathbf{x_{t-1}}$ from $\mathbf{x_t}$ by predicting the denoised observation, which essentially approximates reversing the diffusion process:

$$\mathbf{x}_{t-1} = \sqrt{\alpha_{t-1}} \left( \frac{\mathbf{x}_t - \sqrt{1-\alpha_t}\epsilon_\theta^{(t)}(\mathbf{x}_t)}{\sqrt{\alpha_t}} \right) + \sqrt{1 - \alpha_{t-1} - \sigma_t^2} \cdot \epsilon_\theta^{(t)}(\mathbf{x}_t) + \sigma_t \epsilon_t \tag{35}$$

During the process, DDIM employs an ODE solver to manage the continuous transformation across the latent space:

$$\mathrm{d}\mathbf{x}(t) = \epsilon_\theta^{(t)} \left( \frac{\mathbf{x}(t)}{\sqrt{\sigma^2+1}} \right) \mathrm{d}\sigma(t) \tag{36}$$

Eq.(36) is key to the efficient management of the generation process, allowing for fewer steps in the generative sequence by smoothly interpolating between states using an ODE solver, thus significantly reducing the time complexity compared to traditional methods.

While DDIM's ODE formulation Eq.(12) and its implementation through Eq.(36) provide a foundation for deterministic sampling, Liu et al. (2022a) identifies two critical issues in the ODE formulation of DDIM: first, the neural network $\theta$ and ODE are only well-defined within a narrow data manifold, while numerical methods generate samples outside this region. second, the ODE becomes unbounded as $t \rightarrow 0$ for linear schedules. Therefore PNDM is proposed to decompose the numerical solver into gradient and transfer components. It achieves second-order convergence, enabling 20x speedup while maintaining quality and reducing FID by 0.4 points at the same step count across different datasets and variance schedules.

The DPM-solver (Lu et al., 2022) and Diffusion Exponential Integrator Sampler (DEIS) (Zhang & Chen, 2022) innovate by leveraging the semi-linear structure of the probability flow ODE Eq.(12) to design custom ODE solvers that outperform traditional Runge-Kutta (Hochbruck & Ostermann, 2010) methods in terms of efficiency. Specifically, DPM-solver solves the linear part of the equation and uses neural networks to approximate the nonlinear component. Compared to PNDM, DPM-solver maintains lower FID scores at the same NFE. Further, DEIS employs an Exponential Integrator (Hochbruck & Ostermann, 2010) to discretize ODEs. This method simplifies the probability flow ODE by transforming the probability ODE into a simple non-stiff ODE. Both of the innovations reduce the number of iterations needed producing high-quality samples within just 10 to 20 iterations.

To reduce the computational overhead, Zheng et al. (2023) presents an improved technique for maximum likelihood estimation of ODEs. Instead of directly working with the drift and score terms in Eq.(12), it introduces velocity parameterization to predict and optimize velocity changes $\mathrm{d}\mathbf{x}_t$ during the diffusion process directly. The method improves upon previous ODE-based approaches by incorporating second-order flow matching for more precise trajectory estimation. Additionally, it introduces a negative log-signal-to-noise-ratio (log-SNR) for timing control of the diffusion process, alongside normalized velocity and importance sampling to reduce variance and optimize training. These enhancements significantly improve the model's likelihood estimation performance on image datasets without variational dequantization or data augmentation. While previous methods focus on improving reverse ODE integrators based on Eq.(12), Denoising MCMC (DMCMC) (Kim & Ye, 2022) takes a different approach by integrating Markov Chain Monte Carlo (MCMC) with ODE integrators to optimize the data sampling process. In DMCMC, MCMC first generates initialization points in the product space of data and diffusion time, which are closer to a noise-free state, significantly reducing the noise levels that need to be processed by the ODE integrators. This hybrid approach complements rather than improves the ODE integrators directly, enhancing overall sampling efficiency.

Besides, Lu & Song (2024) focuses on improving continuous-time consistency models(CMs) (Song et al., 2023; Song & Dhariwal, 2023) for efficient diffusion sampling by modifying the ODE parameterization and training objectives of continuous-time CMs. The core contribution is TrigFlow, a unified framework that

bridges EDM (Karras et al., 2022) and Flow Matching (Peluchetti, 2023; Lipman et al., 2022; Liu et al., 2022b; Albergo et al., 2023; Heitz et al., 2023).

While the traditional probability flow framework is governed by Eq.(12), they propose a simplified parameterization. To model these dynamics, they introduce a neural network $\mathbf{F}_\theta$ with parameters $\theta$ that takes normalized samples and time encodings as input. The time variable $t$ is transformed by $c_{noise}(t)$ to better condition the network. This results in a concise probability flow ODE:

$$\frac{\mathrm{d}\mathbf{x}_t}{\mathrm{d}t} = \sigma_d \mathbf{F}_\theta(\frac{\mathbf{x}_t}{\sigma_d}, c_{noise}(t)) \tag{37}$$

By introducing this simplified ODE parameterization, TrigFlow enables training large-scale CMs (up to 1.5B parameters) that achieve state-of-the-art performance with just two sampling steps, significantly reducing computational costs compared to DPM-solver (Lu et al., 2022) and other traditional diffusion models.

Moreover, (Du et al., 2024) replaces SDE with ODE's deterministic trajectory, combined with fixed noise and Consistency Distillation Sampling loss, which minimizes stochastic fluctuations and computational redundancy. These designs enable faster convergence while maintaining high fidelity.

In conclusion, recent research has produced numerous works on faster diffusion samplers based on solving the ODE Eq.(12). Research shows that ODE samplers are highly effective when only a limited number of NFEs is available, while SDE samplers offer better resilience to prior mismatches (Nie et al., 2023) and exhibit superior performance with a greater Number of Function Evaluations (NFEs) (Lu et al., 2022).

### 3.3.2 Sampling Scheduling

In diffusion models, a sampling schedule outlines a structured approach for timing and managing the sampling steps to improve both the efficiency and quality of the model's output. It focuses on optimizing the sequence and timing of these steps, utilizing advanced techniques to process multiple steps simultaneously or in an improved sequential order. Specifically, this scheduling primarily targets the optimization of the reverse process in DDPM, as described in Eq.(4), where each step requires model prediction to gradually denoise from pure noise to the target sample. This scheduling is crucial for reducing computational demands and enhancing the model's performance in generating high-quality samples.

**Parallel Sampling.** Parallel sampling is a process that schedules sampling tasks in parallel. Traditional diffusion models require a extensive series of sequential denoising steps to generate a single sample, which can be quite slow. For instance, Denoising Diffusion Probabilistic Models (DDPMs) (Ho et al., 2020) might need thousands of these steps to produce one sample. However, parallel sampling leverages the power of a multi-core GPU to compute multiple sampling steps. This approach optimizes the use of computational resources and reduces the time needed for model generation. Currently, there is significant work on autoregressive models that employ parallelization to speed up the sampling process.

However, these techniques cannot be directly applied to diffusion models. This is because the computational frameworks and inference efficiency in autoregressive models differ from those in diffusion models. Therefore, designing algorithms tailored to parallelize the sampling process of diffusion models is crucial. An innovative extension of the Denoising Diffusion Implicit Model (DDIM) (Song et al., 2020a) using Deep Equilibrium (DEQ) models is presented (Pokle et al., 2022), where the sampling sequence is conceptualized as a multivariate fixed-point system. This approach focuses on finding the system's fixed point during the forward pass and utilizes implicit differentiation during the backward pass to enhance computational efficiency. By treating the sampling steps as an equilibrium system and solving for their fixed points simultaneously, parallel processing on multiple GPUs is achieved by batching the workload. Notably, it improves efficiency by updating each state $\mathbf{x}_t$ based on predictions from the noise prediction network

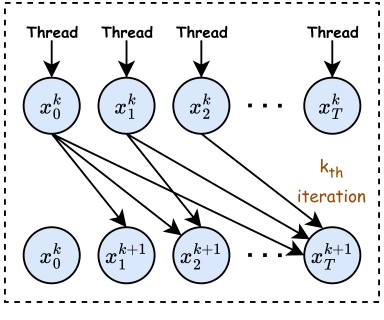

Picard iteration

Figure 15: Computation graph of Picard iterations, which introduces skip dependencies (Shih et al., 2024).

$\epsilon_\theta$, which takes into account all subsequent states $\mathbf{x}_{t+1:T}$, unlike traditional diffusion processes that update states sequentially based only on the immediate next state $\mathbf{x}_{t+1}$.

ParaDiGMS (Shih et al., 2024) employs Picard iterations to parallelize the sampling process in diffusion models. This method models the denoising process using ordinary differential equations (ODEs) (Song et al., 2020c), where Picard iterations approximate the solution to these ODEs concurrently for multiple state updates. ParaDiGMS operates within a sliding window framework, enabling the simultaneous update of multiple state transitions. Each state is iteratively connected to different generations, allowing for information integration from several previous iterations Figure 15.

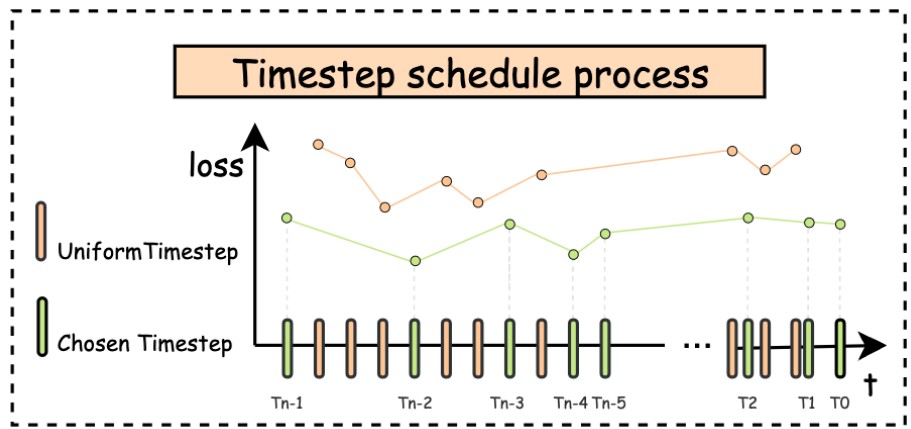

Figure 16: Illustration of timestep schedule optimization process.

Building upon these parallel processing concepts, ParaTAA (Tang et al., 2024) also adopts an iterative approach, primarily applied in practical deployments for image generation tasks such as text-to-image transformations using Stable Diffusion. Specifically, ParaTAA enhances parallel sampling by solving triangular nonlinear equations through fixed-point iteration. Furthermore, the study introduces a novel variant of the Anderson Acceleration (Walker & Ni, 2011) technique, named Triangular Anderson Acceleration, designed to accelerate computation speed and improve the stability of iterative processes. Kodaira et al. (2023) introduces StreamDiffusion for real-time interactive diffusion by combining batched denoising to exploit GPU parallelism and Residual Classifier-Free Guidance(RCFG) to minimize redundant guidance computations, and input-output queues for asynchronous processing. It further optimizes energy efficiency via stochastic similarity filtering, which dynamically skips processing for near-identical frames using cosine similarity thresholds.

**Timestep Schedule.** In the sampling process of diffusion models, the entire process is discrete, and the model progressively restores data from noise through a series of discrete timesteps. Each timestep represents a small denoising step that moves the model from its current state closer to the real data. The timestep schedule refers to the strategy for selecting and arranging these timesteps. It may involve distributing them evenly or performing denser sampling during key stages to ensure the efficiency of the sampling process and the quality of the generated results. Selecting an appropriate method to choose a series of timesteps can enable the sampling process to converge quickly, which the process is shown in Figure 16.

FastDPM (Kong & Ping, 2021) is a unified framework for fast sampling in diffusion models that innovatively generalizes discrete diffusion steps to continuous ones and designs a bijective mapping between these continuous diffusion steps and noise levels. By utilizing this mapping, FastDPM constructs an approximate diffusion and reverse process, significantly reducing the number of steps required ($S \ll T$). It allows for the flexible determination of sampling points by selecting specific steps or variances from the original diffusion process, thereby enhancing efficiency. Watson et al. (2021) proposes a dynamic programming algorithm to optimize timestep scheduling in Denoising Diffusion Probabilistic Models (DDPMs). The algorithm efficiently determines the optimal timestep schedule from thousands of possible steps by leveraging the decomposable

property of Evidence Lower Bound (ELBO) across consecutive timesteps and treating timestep selection as an optimization problem. Experiments show that the optimized schedule requires only 32 timesteps to achieve comparable performance to the original model with thousands of steps, effectively balancing efficiency and quality. However, optimizing an exact Evidence Lower Bound (ELBO) is typically not conducive

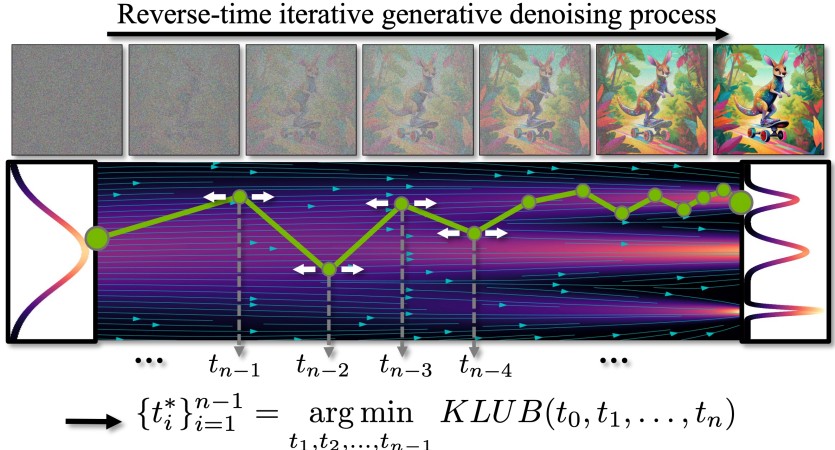

Figure 17: Minimizing an upper bound on the Kullback-Leibler divergence (KLUB) between the true and linearized generative SDEs to find optimal DM sampling schedules (Sabour et al., 2024).

to enhancing image quality. To address this, Align Your Steps (AYS) (Sabour et al., 2024) proposes a compute-efficient paradigm for diffusion models by co-optimizing sampling schedules and solvers. Through stochastic calculus-guided Kullback-Leibler Divergence Upper Bound (KLUB) minimization, as shown in 17, AYS derives adaptive schedules that reduce sampling steps by 40% while maintaining quality—achieving equivalent FID scores with 20 steps versus 30 steps in baseline methods. The optimized schedules are versatile across modalities and solver types, enabling plug-and-play acceleration without model retraining.

### 3.3.3 Truncated Sampling

Truncated sampling enhances the efficiency of sample generation in diffusion models by strategically reducing redundant computations, thereby lowering computational costs. This optimization category encompasses several approaches, with Early Exit and Retrieval-Guided Initialization representing two primary strategies that target different phases of the diffusion process to improve computational efficiency. Specifically, early Exit focuses on terminating unnecessary computations in later stages of the diffusion process when predictions are confident. Conversely, Retrieval-Guided Initialization improves efficiency in the early stages by leveraging retrieved examples to provide a better initialization, effectively bypassing parts of the iterative refinement process. These approaches allocate computation more effectively by focusing resources on the most critical steps of the sampling process.

**Early Exit.** Recent papers that focus on early exit mechanisms have gained significant attention in the Large Language Models (LLMs) domain (Schuster et al., 2021). By implementing early exit strategies, these methods bypass unnecessary computations in deeper layers when outputs can be generated at earlier stages, thereby substantially reducing inference time and computational resources.

Similarly, for computation-intensive diffusion models that typically require numerous denoising steps, early exit techniques present a promising approach to accelerate the generation process dramatically. Moon et al. (2024a) proposes a simple yet effective early exiting framework called Adaptive Score Estimation (ASE) specifically designed to accelerate the sampling process of diffusion models. The method is based on the key observation that score estimation difficulty varies across different timesteps in the diffusion process, with timesteps closer to the noise distribution requiring fewer computational resources than those closer to the data distribution.

The schematic in Figure 18 illustrates the time-dependent exit schedule used in ASE. As shown in the figure, the model progressively skips more building blocks as the diffusion process moves closer to the noise distribution, where the score estimation becomes easier. For example, in the DiT model (Peebles & Xie, 2023), the blocks are dropped progressively as the timestep approaches 1, closer to the noise regime. Conversely, when the timestep is closer to the data regime, more blocks are retained to ensure accurate score estimation. Moreover, the U-ViT model (Bao et al., 2023) follows a similar dropping schedule, but due to the long skip connections between the encoder and decoder, only decoder blocks are skipped. This preserves critical information from the encoder while still speeding up the process.

Through carefully designed time-varying exit schedules, the method significantly accelerates sampling speed while preserving generation quality. It can also be seamlessly integrated with various diffusion model solvers to enhance overall efficiency further.

In contrast, Tang et al. (2023a) introduces DeeDiff, which approaches early exiting through a dynamic uncertainty-aware framework. While ASE relies on static, predefined exit schedules that are fixed during inference, DeeDiff incorporates a timestep-aware uncertainty estimation module (UEM) that adaptively assesses the prediction uncertainty of each intermediate layer at every sampling step. The UEM directly estimates uncer-

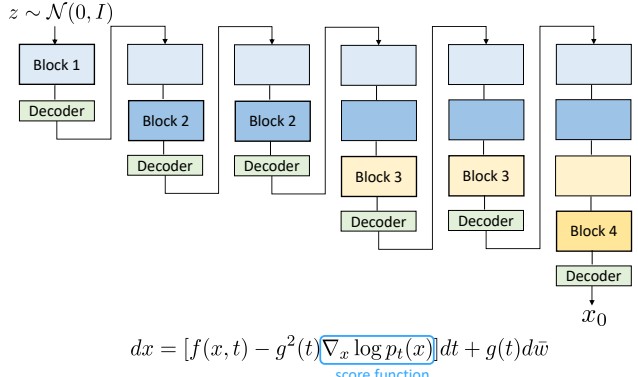

Figure 18: Illustration of the time-dependent exit schedule, where more blocks are skipped as the sampling process moves toward the noise distribution, optimizing computational efficiency Moon et al. (2024a).

tainty values from the features at each layer using lightweight, fully-connected networks, which are trained to indicate how well the current layer's output matches the final layer's prediction. During inference, when a layer's uncertainty falls below a predefined threshold, the model exits early at that layer, bypassing deeper layers for that particular timestep. Despite their different implementation strategies, both methods demonstrate that early exiting frameworks can effectively balance computational efficiency and generation quality, offering practical solutions for deploying large-scale diffusion models.

**Retrieval-Guided Denosing.** Retrieval-Guided Initialization combines the efficiency of retrieval mechanisms with the generative power of diffusion models, and can be applied across various generative tasks. This method guides the generation process by retrieving samples or data relevant to the input. For example, as illustrated in Figure 19, when the input is a text prompt, the retriever selects relevant images from a database, which then serve as contextual guidance for the diffusion model to generate a coherent output. By providing a more informed state for the diffusion model, this approach helps the model converge more quickly and generate higher-quality outputs.

Retrieval-Guided Denoising is widely used in text-to-image diffusion tasks, offering an efficient way to generate images that align with textual descriptions. The kNN-Diffusion (Sheynin et al., 2022) method integrates large-scale retrieval techniques with the generative power of diffusion models, offering an efficient approach for image generation. The key idea is to use the CLIP (Radford et al., 2021) image encoder to map images into a shared embedding space and employ the k-Nearest-Neighbors (kNN) retrieval to identify the k most similar image embeddings. Notably, this method does not require paired text-image datasets, which is a significant advantage for domains where such datasets are scarce. During training, the model is conditioned on image embeddings, and kNN retrieval is used to extend the distribution of conditioning embeddings, which helps the model generalize better and bridge the gap between text and image distributions. In the sampling phase, the model receives text input, which is converted to text embedding using CLIP. The kNN retrieval mechanism is then applied to find the k most similar image embeddings based on the text embedding. These retrieved image embeddings serve as additional conditional information, guiding the generation process and ensuring that the generated image aligns with the textual description. Consequently, kNN-Diffusion im-

proves both the efficiency of image generation and the computational resource consumption, making it a highly efficient solution for text-to-image generation tasks. Unlike KNN-Diffusion, which primarily relies on image retrieval to guide the sampling process, Re-Imagen (Chen et al., 2022b) enhances the generation process by incorporating multimodal retrieval to balance computational cost and output quality, using both image and text pairs. Re-Imagen's retrieval method is based on an external multimodal knowledge base. During sampling, given an input text, the model queries an external database to retrieve the most relevant image-text pairs. These retrieved pairs are then used as additional conditional inputs for the generation process. To identify the most relevant pairs, the model employs either BM25 (Robertson et al., 2009) or CLIP (Radford et al., 2021) similarity measures, which evaluate the alignment between the input text and the image-text pairs in the database. Therefore, this approach provides more diverse visual information, especially for rare or unseen entities. Additionally, Re-Imagen employs a cascaded diffusion architecture (Ho et al., 2022a), which allows it to generate high-quality images more efficiently compared to kNN-Diffusion, as it reduces the computational cost by progressively refining images at different resolutions.

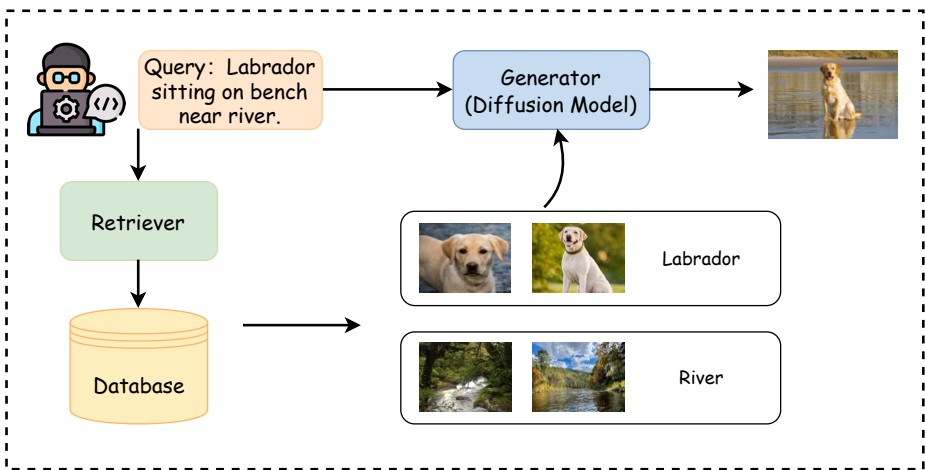

Figure 19: Illustration of the retrieval-based diffusion model. The retriever selects relevant images from a database based on input text. These retrieved images provide contextual guidance for the generator (diffusion model) to produce a new, coherent output image.

Besides approaches that rely on similarity measures like CLIP embeddings to retrieve images or texts, ReDi (Retrieval-based Diffusion) (Zhang et al., 2023c) offers a fundamentally different approach to accelerating diffusion model inference. ReDi directly retrieves precomputed trajectories from the diffusion process itself. The method constructs a knowledge base of sample trajectories where each entry contains a key-value pair. For instance, in the forward diffusion process (noise-adding process), an early state sample $x_k$ serves as the key, while a later state sample $x_v$ serves as the value. During inference, ReDi first generates the initial steps of a trajectory up to $x'_k$, uses this as a query to find a similar state $x_k$ in the knowledge base, and then skips intermediate computational steps by jumping to the retrieved $x_v$ before continuing the generation process. By skipping a large portion of intermediate steps, ReDi significantly reduces the number of function estimations (NFEs) required during sampling. Experiments with Stable Diffusion demonstrate that ReDi achieves a two times speedup with comparable quality and enables effective zero-shot domain adaptation for tasks like image stylization without requiring domain-specific knowledge bases.

### 3.3.4 Knowledge Distillation

Knowledge distillation (Hinton et al., 2015) is a technique that compresses complex models into smaller, efficient versions with minimal performance loss. The process of knowledge distillation can be captured by minimizing the following loss function:

$$L_{\text{KD}} = \alpha L_{\text{CE}}(y, \sigma(T_s(x))) + \beta L_{\text{MSE}}(T_t(x), T_s(x)), \tag{38}$$

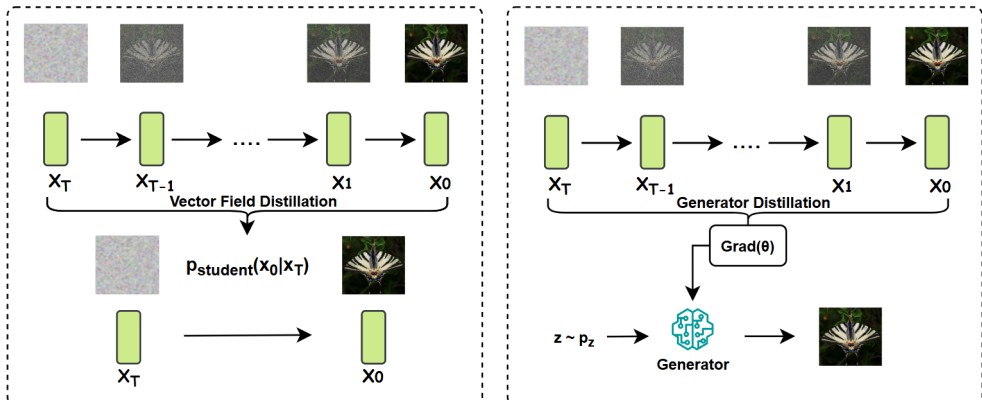

Figure 20: Illustrations of the knowledge distillation.

where $T_t$ and $T_s$ are the teacher and student models, respectively, $\sigma$ is the softmax function, $L_{\text{CE}}$ is the cross-entropy loss, and $L_{\text{MSE}}$ is the mean squared error loss, with $\alpha$ and $\beta$ as balancing hyperparameters. In DMs, known for generating high-quality data, this approach is increasingly applied to improve efficiency by addressing slow sampling speeds caused by the numerous neural function evaluations in the diffusion process. By distilling the knowledge from DMs into more efficient forms, researchers aim to accelerate sampling while preserving the generative performance of the original models. Follow Luo (2023), knowledge distillation for DMs can be categorized into vector field distillation and generator distillation.

**Vector Field Distillation.** Vector field distillation improves the efficiency of deterministic sampling in diffusion models by transforming the generative ODE into a new generative vector field. This approach reduces the number of NFEs needed to produce samples of similar quality. Luhman & Luhman (2021) first proposes a strategy to distill a DDIM sampler into a Gaussian model that needs only one NFE for sampling. In this approach, a conditional Gaussian model serves as the student model, and the training process involves minimizing the conditional KL divergence between this student model and the DDIM sampler. While this method advances the application of knowledge distillation to diffusion models, it still has computational inefficiencies, as it necessitates generating the final outputs of DDIM or other ODE samplers, which entails hundreds of NFEs for each training batch. Salimans & Ho (2022) proposes a progressive distillation strategy to train a student model to use half the NFEs of the teacher model by learning its two-step prediction strategy, as illustrated in Figure 21. Once the student model accurately predicts the teacher's two-step sampling strategy, it replaces the teacher model, and a new student model is trained to further reduce the sampling steps. This method reduces the NFEs significantly, achieving 250 times greater efficiency with only a 5% drop in performance.

A two-stage distillation strategy is proposed by Meng et al. (2023) to address the challenge of transferring knowledge from classifier-free guided conditional diffusion models like DALL · E-2 (Ramesh et al., 2022) and Stable Diffusion (Rombach et al., 2022). In the first stage, a student model is trained with classifier-free guidance to learn from the teacher diffusion model. The second stage employs the progressive diffusion strategy to further reduce the number of diffusion steps for the student model. This two-stage approach is applied to both pixel-space and latent-space models for various tasks, including text-guided generation and image inpainting.

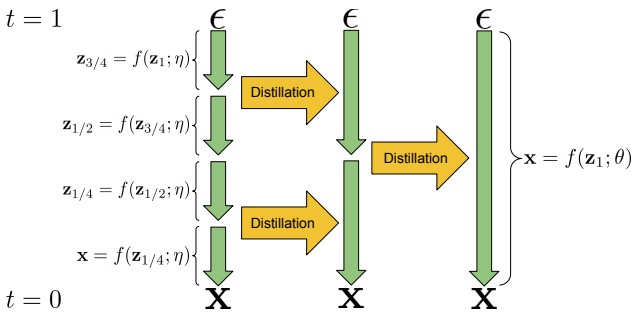

Figure 21: The progressive distillation, where the original sampler derived from integrating a learned diffusion model's probability flow ODE, is efficiently condensed into a new sampler that achieves the same task in fewer steps. (Salimans & Ho, 2022).

Song et al. (2023) firtstly introduces the Consistency Model (CM), which leverages the self-consistency property of generative ODEs in diffusion models. Instead of directly mimicking the output of the generative ODE, their method focuses on minimizing the difference in the self-consistency function. By randomly diffusing a real data sample and simulating a few steps of the generative ODE to generate another noisy sample on the same ODE path, the model inputs these two noisy samples into a student model. Consequently, Chen et al. (2025) accelerates consistency models via hybrid distillation, enabling 1-4 step generation. It transforms pre-trained flow models into TrigFlow (Lu & Song, 2024) without retraining, preserving trajectory alignment while boosting fidelity via adversarial training LADD. Wu et al. (2024) proposes Multi-scale Latent

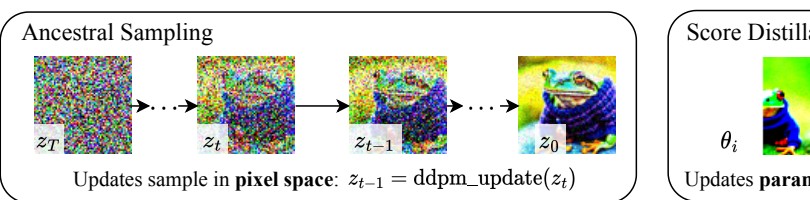 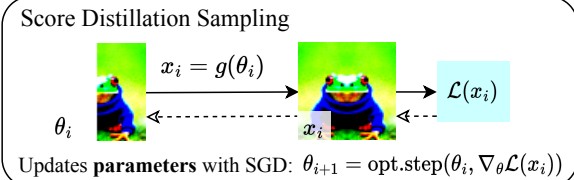

Figure 22: Illustration as (Poole et al., 2022), it utilizes score distillation sampling.

Point Consistency Model (MLPCM), which implements one-step generation through consistency distillation, combines multi-scale latent space and 3D attention mechanisms to reduce computational complexity. Moreover, Yin et al. (2024) proposes Distribution Matching Distillation (DMD), which distills multi-step diffusion models into a single-step generator by introducing a distribution-level matching objective that minimizes the KL divergence between real and synthetic data distributions, alongside a regression loss to align large-scale structural features.

**Generator Distillation.** Unlike vector field distillation, which primarily focuses on distilling knowledge into student models with identical input and output dimensions, generator distillation aims to transfer the complex distributional knowledge embedded in a diffusion model into a more efficient generator. The Neural Radiance Field (NeRF) (Mildenhall et al., 2021) is a powerful technique for reconstructing 3D scenes from 2D images by learning a continuous volumetric scene representation. NeRFs generate photorealistic views of scenes from novel angles, making them valuable for applications in computer vision and graphics.

However, the limited availability of data for constructing NeRFs is an issue. Therefore, exploring distillation methods to obtain NeRFs with contents related to given text prompts is a promising way. (Poole et al., 2022) first proposed Score Distillation Sampling (SDS) to distill a 2D text-to-image diffusion model into 3D NeRFs, as illustrated in Figure 22. Unlike traditional NeRF construction that requires images from multiple views of the target 3D objects, text-driven construction of NeRF lacks both the 3D object and the multiple views. The SDS method optimizes the NeRF by minimizing the diffusion model's loss function using NeRF-rendered images from a fixed view.

Wang et al. (2024b) introduce Variational Score Distillation (VSD), which extends SDS by treating the 3D scene corresponding to a textual prompt as a distribution rather than a single point. Compared to SDS, which generates a single 3D scene and often suffers from limited diversity and fidelity, VSD is capable of generating more varied and realistic 3D scenes, even with a single particle. Luo et al. (2024) propose Diff-Instruct, which can transfer knowledge from pre-trained diffusion models to a wide range of generative models, all without requiring additional data. The key innovation in Diff-Instruct is the introduction of Integral Kullback-Leibler divergence, which is specifically designed to handle the diffusion process and offers a more robust way to compare distributions. Decatur et al. (2024) present Cascaded Score Distillation (CSD), an advancement by addressing a key limitation of standard SDS. Specifically, while traditional SDS only leverages the initial low-resolution stage of a cascaded model, CSD distills scores across multiple resolutions in a cascaded manner, allowing for nuanced control over both fine details and the global structure of the supervision. By formulating a distillation loss that integrates all cascaded stages, which are trained independently, CSD enhances the overall capability of generating high-quality 3D representations.

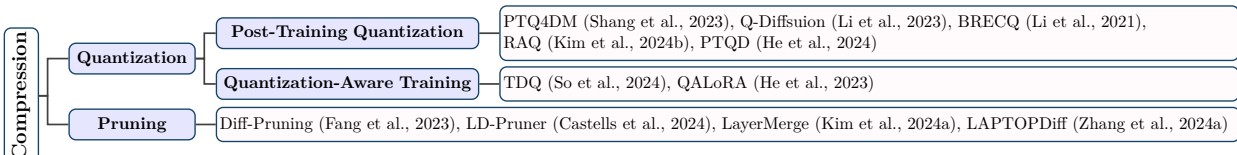

Figure 23: Summary of compression techniques for DMs.

## 3.4 Compression

Model compression enhances efficiency by reducing the sizes and the amount of arithmetic operations of DM. As summarized in Figure 23, model compression techniques for DMs can be grouped into quantization and pruning. These two categories are orthogonal to each other, and compress DMs from different perspectives.

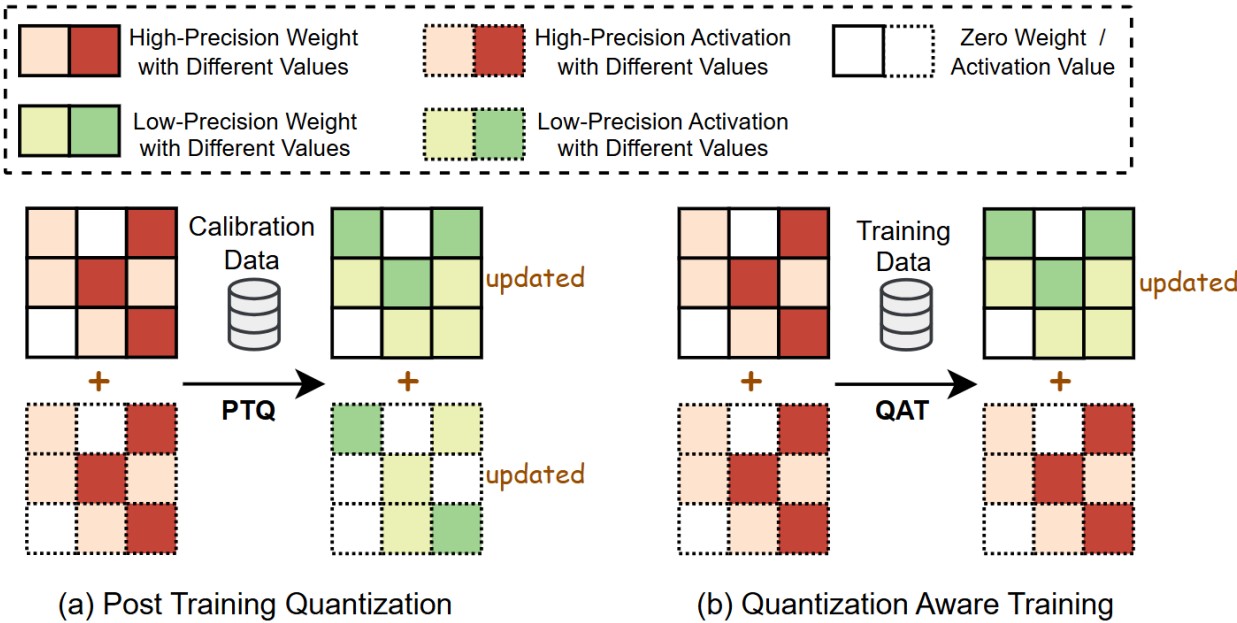

Figure 24: Illustrations of the quantization.

### 3.4.1 Quantization

Quantization compresses neural networks by converting model weights and/or activations of high-precision data types $\mathbf{X}^H$ such as 32-bit floating point into low-precision data types $\mathbf{X}^L$ such as 8-bit integer (Dettmers et al., 2024). Quantization techniques can be classified into post-training quantization (PTQ) and quantization-aware training (QAT).

**Post-Training Quantization.** PTQ involves selecting operations for quantization, collecting calibration samples, and determining quantization parameters for weights and activations. While collecting calibration samples is straightforward for CNNs and ViTs using real training data, it poses a challenge for Diffusion Models (DMs). In DMs, the inputs are generated samples $\mathbf{x}_t$ at various time steps (t = 0, 1, ..., T), where T is large to ensure convergence to an isotropic Normal distribution. To address this issue, Shang et al. (2023) proposes PTQ4DM, the first DM-specific calibration set collection method, generating calibration data across all time steps with a specific distribution. However, their explorations remain confined to lower resolutions and 8-bit precision. Q-Diffsuion (Li et al., 2023) propose a time step-aware calibration data sampling to improve calibration quality and apply BRECQ (Li et al., 2021), which is a commonly utilized PTQ framework, to improve performance. Furthermore, compared to conventional PTQ calibration methods, they identify the accumulation of quantization error across time steps as another challenge in quantizing DMs Figure 25

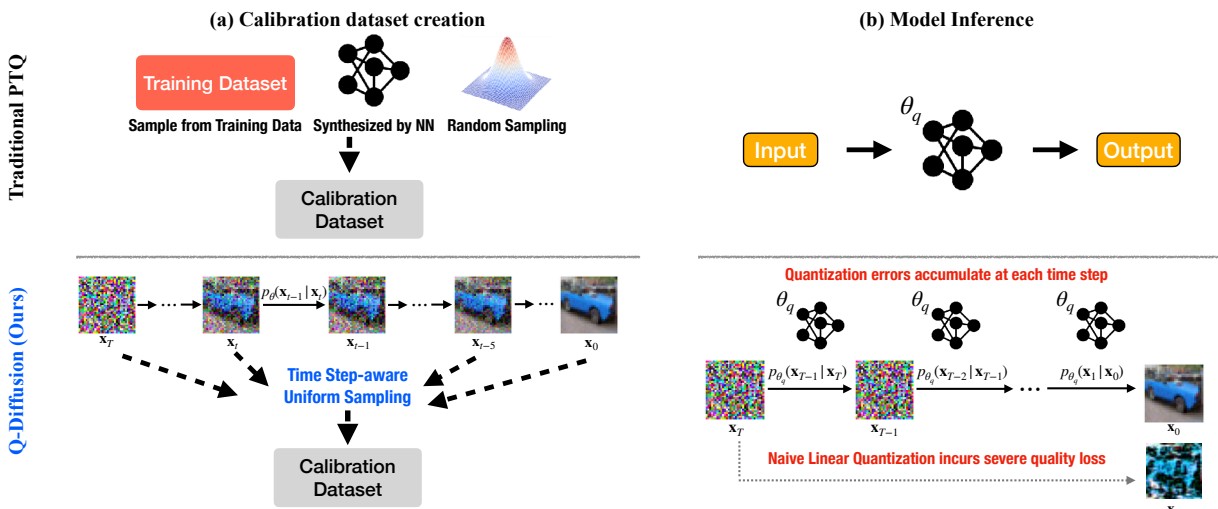

Figure 25: Traditional PTQ scenarios and Q-Diffusion differ in (a) the creation of calibration datasets and (b) the workflow for model inference (Li et al., 2023).

(a). Therefore, they also propose a specialized quantizer for the noise estimation network shown in Figure 25 (b). Based on Q-Diffusion, Kim et al. (2024b) find that inaccurate computation during the early stage of the reverse diffusion process has minimal impact on the quality of generated images. Therefore, they introduce a method that focuses on further reducing the number of activation bits for the early reverse diffusion process while maintaining high-bit activations for the later stages. Lastly, He et al. (2024) presents PTQD, a unified formulation for quantization noise and diffusion perturbed noise. Additionally, they introduce a step-aware mixed precision scheme, which dynamically selects the appropriate bitwidths for synonymous steps.

**Quantization-Aware Training.** Different from PTQ, QAT quantizes diffusion models during the training process, allowing models to learn quantization-friendly representations. Since QAT requires additional training after introducing quantization operators, it is much more expensive and time-consuming than PTQ. So et al. (2024) proposes a novel quantization method that enhances

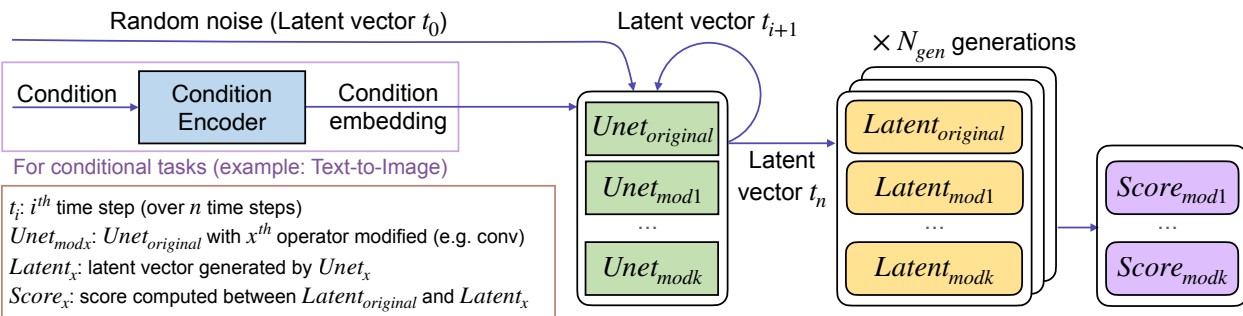

Figure 26: Pruning evaluates changes in the central tendency and variability to determine the significance of each operator. (Castells et al., 2024).

output quality by dynamically adjusting the quantization interval based on time step information. The proposed approach integrates with the Learned Step Size Quantization (Esser et al., 2019) framework, replacing the static quantization interval with a dynamically generated output from the Time-Dynamic Quantization module. This dynamic adjustment leads to significant improvements in the quality of the quantized outputs. He et al. (2023) introduces a quantization-aware low-rank adapter that integrates with model weights and is jointly quantized to a low bit-width. This

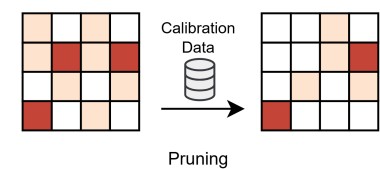

Figure 27: Illustrations of the pruning.

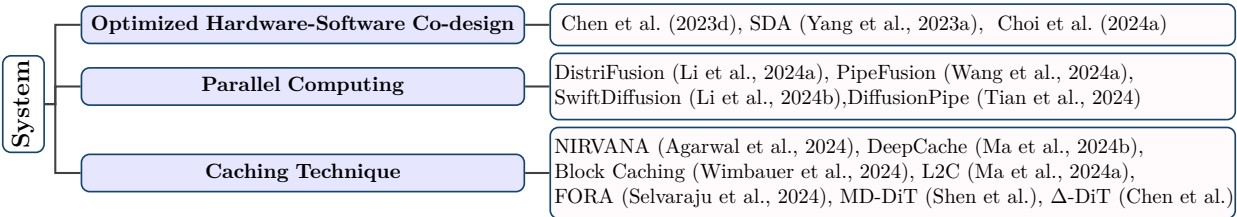

Figure 28: Summary of system-level efficiency optimization techniques for diffusion models.

approach distills the denoising capabilities of full-precision models into their quantized versions, utilizing only a few trainable quantization scales per layer and eliminating the need for training data.

### 3.4.2 Pruning

Pruning compresses DMs by removing redundant or less important model weights. Currently, most pruning methods for DMs focus on pruning structured patterns such as groups of consecutive parameters or hierarchical structures. For instance, Diff-Pruning (Fang et al., 2023) introduces the first dedicated method designed for pruning diffusion models. Diff-Pruning leverages Taylor expansion over pruned timesteps to estimate the importance of weights. By filtering out non-contributory diffusion steps and aggregating informative gradients, Diff-Pruning enhances model efficiency while preserving essential features. LD-Pruner (Castells et al., 2024), as illustrated in Figure 26, on the other hand, proposes a pruning method specifically designed for Latent Diffusion Models (LDMs) The key innovation of LD-Pruner lies in its utilization of the latent space to guide the pruning process. The method enables a precise assessment of pruning impacts by generating multiple sets of latent vectors—one set for the original Unet and additional sets for each modified Unet where a single operator is altered. The importance of each operator is then quantified using a specialized formula that considers shifts in both the central tendency and variability of the latent vectors. This approach ensures that the pruning process preserves model performance while adapting to the specific characteristics of LDMs.

Kim et al. (2024a) introduces a technique known as LayerMerge, designed to jointly prune convolution layers and activation functions to achieve a desired inference speedup while minimizing performance degradation. LayerMerge addresses the challenge of selecting which layers to remove by formulating a new surrogate optimization problem. Given the exponential nature of the selection space, the authors propose an efficient solution using dynamic programming. Their approach involves constructing dynamic programming (DP) lookup tables that exploit the problem's inherent structure, thereby allowing for an exact and efficient solution to the pruning problem.

Lastly, LAPTOPDiff (Zhang et al., 2024a) introduces a layer-pruning technique aimed at automatically compressing the U-Net architecture of diffusion models. The core of this approach is an effective one-shot pruning criterion, distinguished by its favorable additivity property. This property ensures that the one-shot performance of the pruning is superior to other traditional layer pruning methods and manual layer removal techniques. By framing the pruning problem within the context of combinatorial optimization, LAPTOPDiff simplifies the pruning process while achieving significant performance gains. The proposed method stands out for its ability to provide a robust one-shot pruning solution, offering a clear advantage in compressing diffusion models efficiently.

## 4 System-Level Efficiency Optimization

### 4.1 Hardware-Software Co-Design

The co-design of hardware and software is pivotal for achieving efficient deployment of diffusion models in real-time and resource-constrained environments. Following algorithm-level optimizations, system-level techniques focus on integrating hardware-specific features, distributed computation, and caching mechanisms. These strategies aim to address the computational complexity and memory demands of large-scale diffusion

models, enabling more practical applications across various platforms like GPUs, FPGAs, and mobile devices. One significant contribution is the work by Chen et al. (2023d), which explores GPU-aware optimizations for accelerating diffusion models directly on mobile devices. Implementing specialized kernels and optimized softmax operations reduces inference latency, achieving near real-time performance on mobile GPUs.

In a related effort, Yang et al. (2023a) propose SDA, a low-bit stable diffusion accelerator designed specifically for edge FP-GAs. Utilizing quantization-aware training and a hybrid systolic array architecture as illustrated in Figure 29, SDA effectively balances computational efficiency with flexibility, handling both convolutional and attention operations efficiently. Through a two-level pipelining structure, the nonlinear operators are efficiently integrated with the hybridSA, enabling coordinated operation that enhances processing speed while reducing resource usage. Finally, SDA achieves a speedup of 97.3x when compared to ARM Cortex-A53 CPU. Furthermore, Choi et al. (2024a) introduces a stable diffusion processor optimized

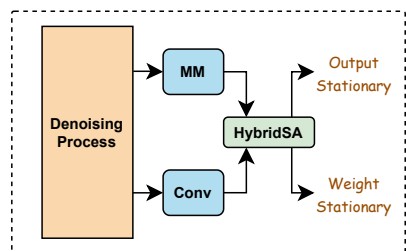

Figure 29: Illustration of the HybridSA architecture from Yang et al. (2023a).

for mobile platforms through patch similarity-based sparsity, mixed-precision strategies and and a Dual-mode Bit-Slice Core (DBSC) architecture that supports mixed-precision computation, which particularly targeting resource-constrained devices such as mobile platforms. Together, these optimizations significantly improve throughput and energy efficiency, making Stable Diffusion more viable for energy-sensitive applications.

For GPU-accelerated mobile applications requiring real-time interactivity, Chen et al. (2023d) provides kernel-level optimizations and mobile-specific operator tuning with strongest latency. When targeting embedded edge devices with strict power budgets and static workloads, Yang et al. (2023a) achieves superior performance via hardware-algorithm synergy. Meanwhile, Choi et al. (2024a) firstly approach dynamic energy constraints in always-on scenarios through runtime sparsity adaptation.

## 4.2 Parallel Computing

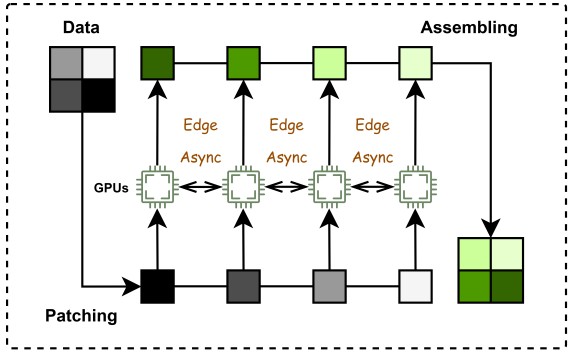
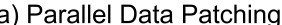

(a) Parallel Data Patching

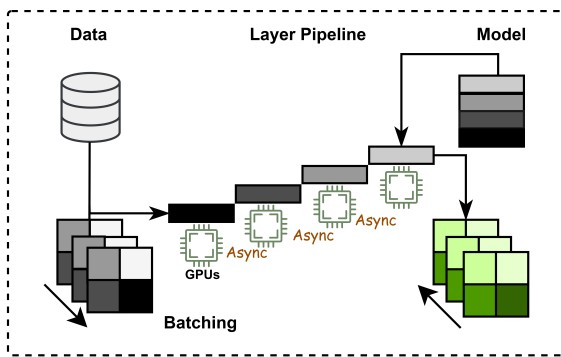

(b) Model Layer Parallel Pipeline

Figure 30: Illustrations of the parallel computing for diffusion models.

Parallel computing Figure 30 plays a critical role in the efficient execution of diffusion models, especially given the computation-intensive nature of these algorithms. Recent advances in parallel computing strategies have enabled significant improvements in inference speed and scalability, often without compromising the quality of the generated output (Li et al., 2024a; Wang et al., 2024a; Li et al., 2024b; Tian et al., 2024). This section highlights several notable contributions that tackle the challenge of parallelizing diffusion models across multiple GPUs and other distributed architectures.

Li et al. (2024a) introduced DistriFusion, a framework designed for distributed parallel inference tailored to high-resolution diffusion models such as SDXL. Their approach involves partitioning the model inputs into distinct patches, which are then processed independently across multiple GPUs. This method leverages

the available hardware resources more effectively, achieving a 6.1x speedup on 8xA100 GPUs compared to single-card operation, all while maintaining output quality. To address potential issues arising from the loss of inter-patch interaction, which could compromise global consistency, DistriFusion employs dynamic synchronization of activation displacements, striking a balance between preserving coherence and minimizing communication overhead.

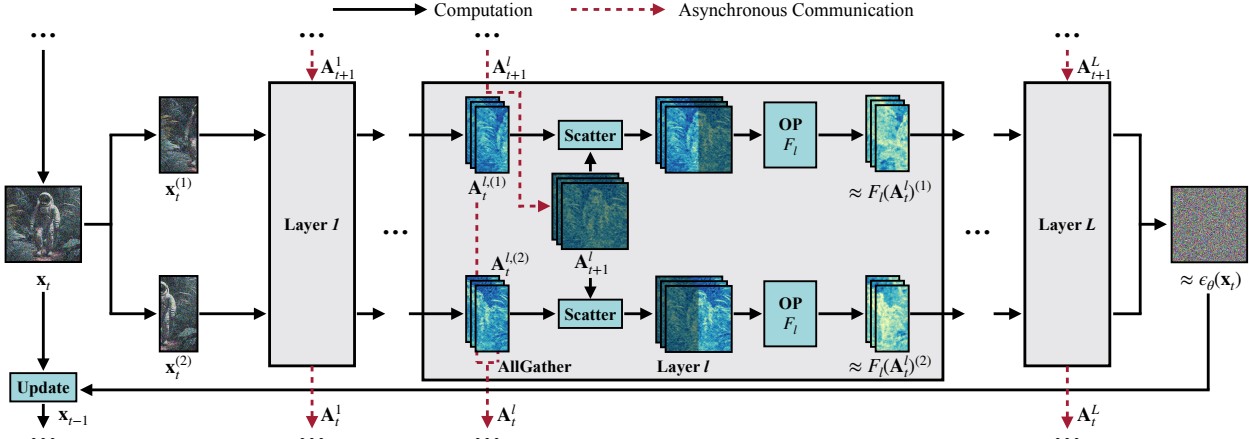

Figure 31: Illustrations of the diffusion architecture from (Li et al., 2024a).

Building on the insights gained from DistriFusion, Wang et al. (2024a) further refined the distributed inference paradigm with PipeFusion. This system not only splits images into patches but also distributes the network layers across different devices, thereby reducing the associated communication costs and enabling the use of PCIe-linked GPUs instead of NVLink-connected ones. PipeFusion integrates sequence parallelism, tensor parallelism, displaced patch parallelism, and displaced patch pipeline parallelism, optimizing workflow for a wider range of hardware configurations. For applications involving add-on modules such as ControlNet and LoRA, Li et al. (2024b) developed SwiftDiffusion, as illustrated in Figure 31. This framework optimizes the serving workflow of these modules, allowing them to run in parallel on multiple GPUs. As a result, SwiftDiffusion delivers a 5x reduction in inference latency and a 2x improvement in throughput, ensuring that enhanced speed does not come at the expense of output quality. Lastly, Tian et al. (2024) focused on the training phase with DiffusionPipe, demonstrating that pipeline parallelism can produce a 1.41x training speedup, while data parallelism contributes an additional 1.28x acceleration. Although the optimization methods for DiffusionPipe were not detailed in the notes, the combination of these parallelization strategies offers a promising direction to improve the efficiency of both the training and inference pipelines for diffusion models. These methodologies encompass a hierarchical optimization, progressing from coarse-grained spatial partitioning to fine-grained module-specific adaptations, thereby empowering practitioners to strategically align parallelism configurations with underlying hardware constraints.

### 4.3 Caching Technique

In diffusion models, the computational hotspot often centers around discrete time-step diffusion, which is characterized by strong temporal locality. Consequently, building an efficient caching system for diffusion models is nonnegligible to enhance its performance. Indeed, extensive research has been conducted on optimizing caching systems in Figure 32, resulting in significant advancements in this field.

Agarwal et al. (2024) proposed NIRVANA, a novel system designed to enhance the efficiency of text-to-image generation using diffusion models. Specifically, the key innovation lies in its approximate caching technique, which reduces computational costs and latency by reusing intermediate noise states from previous image generation processes. Instead of starting from scratch with every new text prompt, NIRVANA retrieves and reconditions these cached states, allowing it to skip several initial denoising steps. Additionally, the system uses a custom cache management policy called Least Computationally Beneficial and Frequently Used (LCBFU), which optimizes the storage and reuse of cached states to maximize computational efficiency. This

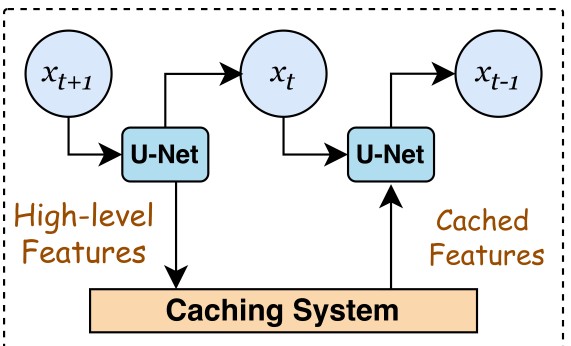 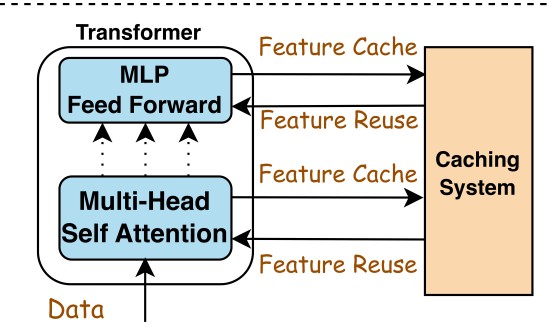

Figure 32: Illustrations of the caching system for diffusion models focus on the U-Net block and the Transformer layer, critical components for effectively implementing caching techniques.

makes NIRVANA particularly suited for large-scale, production-level deployments of text-to-image diffusion models. From another perspective, Ma et al. (2024b) introduces an innovative approach called DeepCache, designed to accelerate the image generation process by leveraging the temporal redundancy in the denoising steps of diffusion models, without the need for additional model training, as illustrated in Figure 33. The key insight is the observation that high-level features, such as the main structure and shape of an image, exhibit minimal changes between adjacent denoising steps. These features can be cached and reused in subsequent steps, thereby avoiding redundant computations. This method takes advantage of the U-Net architecture by combining these cached high-level features with low-level features, updating only the low-level features to reduce computational load, leading to a significant acceleration in the overall process. Wimbauer et al. (2024) proposed Block Caching, a technique that identifies and caches redundant computations within the model's layers during the denoising process. By reusing these cached outputs in subsequent timesteps, the method significantly speeds up inference while maintaining image quality. To optimize this caching process, they introduce an Automatic Cache Scheduling mechanism, which dynamically determines when and where to cache based on the relative changes in layer outputs over time. Additionally, the paper addresses potential misalignment issues from aggressive caching by implementing a Scale-Shift Adjustment mechanism, which fine-tunes cached outputs to align with the model's expectations, thereby preventing visual artifacts.

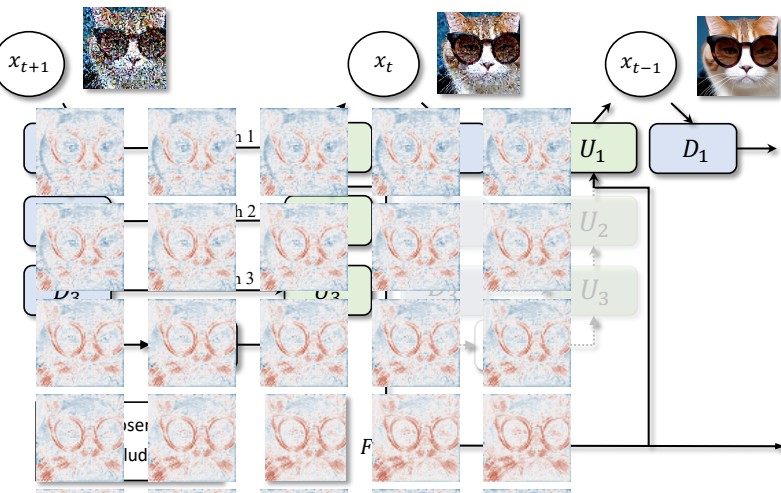

Figure 33: Illustration of the caching system from (Ma et al., 2024b).

Recently, the application of diffusion with transformer models has yielded considerable success. Ma et al. (2024a) is concerned with the introduction of a layer caching mechanissm, designated Learning-to-Cache (L2C), to accelerate diffusion transformer models. L2C exploits the redundancy between layers within the transformer architecture, dynamically caching computations from certain layers to reduce redundant calcu-

lations and lower inference costs. The implementation entails transforming the layer selection problem into a differentiable optimization problem, using interpolation to determine whether to perform a full computation or utilize cached results at different timesteps during inference. In contrast to the emphasis on layer caching, Selvaraju et al. (2024) proposed Fast-Forward Caching (FORA), a technique designed to accelerate Diffusion Transformers (DiT) by reducing redundant computations during the inference phase. The key insight behind FORA is the observation that the outputs from the self-attention and MLP layers in a Transformer exhibit high similarity across consecutive time steps in the diffusion process. To leverage this, FORA implements a static caching mechanism where these layer outputs are cached at regular intervals, which are determined by $N$, and reused for a set number of subsequent steps, thereby avoiding recomputing similar outputs.

Shen et al. introduces MD-DiT, a unified framework for efficient diffusion transformers by integrating block skipping and caching strategies, enabling dynamic depth adjustment across timesteps without additional training. It optimizes computation by caching incremental changes from previous timesteps and selectively skipping non-critical blocks. Chen et al. proposes $\Delta$-DiT, a groundbreaking framework for optimizing diffusion transformers through two synergistic innovations: step-sensitive block caching and intelligently allocated computational resources. By exploiting the temporal consistency inherent in diffusion trajectories, the system strategically reuses intermediate residual features across sequential steps. Notably, it harmonizes both computation-skipping and feature-caching mechanisms within a cohesive optimization architecture.

Generally, these methods demonstrate an trajectory from coarse temporal reuse to fine-grained layer adaptation, guiding practitioners to employ noise-state caching for prompt-variant scenarios, feature-level caching for structural consistency, and hybrid static-dynamic strategies for transformer-based models.

## 5 Frameworks

Table 2: Comparison of Optimization Support in Selected Diffusion Model Frameworks and Techniques. In this table, 'Training' indicates whether the framework accelerates the training process of diffusion models, while 'Inference' indicates whether it accelerates the generation process.

| Framework | Training | Inference | Key Features |
|---|---|---|---|
| Flash Attention | ✓ | ✓ | High-efficiency attention computation for Diffusion Transformers (DiT) |
| xFormers | ✓ | ✓ | Memory-efficient attention and modular ops tailored for diffusion Transformer speedups |
| DeepSpeed | ✓ | ✓ | Scalable distributed training and inference optimizations for large diffusion models |
| OneFlow | ✓ | ✓ | Compiler-optimized pipeline for faster diffusion model training and sampling |
| Stable-Fast | ✗ | ✓ | Fast inference optimization for Diffusers with CUDNN fusion |
| Onediff | ✗ | ✓ | Diffusion-specific acceleration with Deep-Cache and quantization |
| DeepCache | ✗ | ✓ | Reuses cached diffusion features to speed up inference iterations |
| TGATE | ✗ | ✓ | Temporal gating to streamline cross-attention in diffusion inference |
| xDiT | ✗ | ✓ | Parallel inference engine for Diffusion Transformers |

Frameworks in the efficient diffusion landscape refer to high-performance tools and libraries designed to optimize training and/or inference. DM frameworks can be in general grouped based on whether they

support the tasks of training and inference. Specifically, frameworks that support training aim to provide scalable, efficient, and flexible infrastructure that improves computation efficiency, reduces memory footprint, optimizes communication efficiency, and ensures reliability of the training process. Frameworks that support inference focus on optimizing inference throughput and reducing memory footprint and latency.

A subset of frameworks, including Flash Attention (Dao et al., 2022), xFormers (Lefaudeux et al., 2022), DeepSpeed (Rasley et al., 2020), and OneFlow (Yuan et al., 2021), distinguishes itself by offering comprehensive support for both training and inference phases of diffusion models. These tools, rooted in broader deep learning optimization efforts, cater to the full lifecycle of model development. Flash Attention accelerates both training and inference by optimizing attention computation, reducing memory usage and latency through techniques like tiling and recomputation, making it particularly effective for Diffusion Transformers (DiT). xFormers provides memory-efficient Transformer optimizations with a modular design, enabling flexible and efficient computation across a range of resource-intensive tasks. DeepSpeed, originally designed for large-scale model training, extends its distributed capabilities to inference, enabling scalable deployment across multiple GPUs. Similarly, OneFlow leverages its compiler-driven architecture to streamline both training and inference workflows, appealing to researchers and practitioners seeking end-to-end optimization. Together, these frameworks provide robust foundations for advancing diffusion model research and deployment, balancing flexibility with high performance.

In contrast, a growing array of tools—Stable-Fast (chengzeyi, 2024), Onediff (Contributors, 2022), Deep-Cache (Ma et al., 2024b), TGATE (Zhang et al., 2024b; Liu et al.), and xDiT (Fang et al., 2024a;b; Fang & Zhao, 2024; Sun et al., 2024)—focuses exclusively on accelerating the inference stage of diffusion models, addressing the demand for rapid and resource-efficient generation. Stable-Fast optimizes the Hugging Face Diffusers ecosystem, prioritizing low-latency inference for real-time applications. Onediff builds on this trend by integrating cutting-edge techniques like caching and quantization, tailoring its acceleration to diffusion-specific workloads. DeepCache and TGATE introduce innovative caching strategies, exploiting temporal redundancies to reduce computational overhead in inference, particularly for U-Net-based models. Meanwhile, xDiT targets the emerging Diffusion Transformer (DiT) architecture, employing parallelization to enhance inference scalability. These inference-centric tools reflect a shift toward specialized optimizations, catering to the practical needs of deployment in constrained environments or novel model paradigms.

## 6 Future Work

Despite the significant progress made in efficient diffusion models, several promising research directions remain open. In particular, we identify the following key areas for future work:

**Hybridizing Diffusion and Autoregressive Models.** One promising future direction is to explore hybridizing autoregressive and diffusion models to combine the strengths of both paradigms. This integration enables the use of key-value (KV) caching, a technique from autoregressive transformers, to accelerate diffusion and support streamable generation. A representative work is Block Diffusion(Arriola et al., 2025), which segments generation into blocks and applies autoregressive-style caching across denoising steps. This approach opens up promising opportunities to unify architectural benefits, allowing diffusion models to inherit the efficiency and online capabilities of autoregressive methods.

**Without Classifier-Free Guidance (CFG).** Although classifier-free guidance has been widely adopted to enhance generation quality, it introduces substantial computational overhead. Tang et al. (Tang et al., 2025) propose Model-guidance (MG), a novel training objective that eliminates the need for CFG by directly incorporating the posterior probability of conditions rather than solely modeling data distribution. This approach not only doubles inference speed by avoiding the second network forward pass required by CFG but also significantly accelerates model training with $6.5\times$ faster convergence and approximately 60% performance improvement. Notably, when compared to concurrent methods, MG achieves state-of-the-art performance on ImageNet 256 with an FID of 1.34 while requiring only about 12% of the computational resources of comparable approaches. The effectiveness of MG suggests a promising direction for future research to explore alternative guidance mechanisms or fundamentally rethink the training paradigm to better balance computational efficiency and generation quality.

**Efficient Attention Mechanisms for Video Diffusion.** Bidirectional attention has become a dominant component in diffusion architectures, but it introduces substantial computational overhead that scales quadratically with sequence length. This is particularly problematic for video diffusion models where the sequence length grows linearly with the number of frames. Recent works have begun to explore efficient attention mechanisms specifically designed for diffusion transformers (DiTs). Xia et al. (Xia et al., 2025) propose AdaSpa, which leverages the hierarchical sparsity inherent in DiTs through a blockified pattern approach and adaptive search methods. Their work demonstrates that sparse characteristics of DiTs exhibit hierarchical structures between different modalities and remain invariant across denoising steps, enabling significant computational savings. Similarly, Ding et al. (Ding et al., 2025) identify tile-style repetitive patterns in 3D attention maps for video data, introducing sparse 3D attention with linear complexity relative to frame count. Their approach combines efficient attention with consistency distillation techniques to enable up to 7.8× faster generation for high-resolution videos. These developments suggest promising directions for mitigating the computational burden of attention in diffusion models without sacrificing generation quality.

## 7    Conclusion

In this survey, we provide a systematic review of efficient diffusion models, an important area of research aimed at democratizing diffusion models. We start with motivating the necessity for efficient diffusion models. Guided by a taxonomy, we review efficient techniques for diffusion models from algorithm-level and system-level perspectives respectively. Furthermore, we review diffusion models frameworks with specific optimizations and features crucial for efficient diffusion models. We believe that efficiency will play an increasingly important role in diffusion models and diffusion models-oriented systems. We hope this survey could enable researchers and practitioners to quickly get started in this field and act as a catalyst to inspire new research on efficient diffusion models.

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
