# OpenReview forum: "Efficient Diffusion Models: A Survey"
_TMLR — Accepted by TMLR_

### Review · Reviewer_TFC6 · 2025-02-25

**Summary Of Contributions:**

This survey comprehensively reviewed a large number of papers related to efficient diffusion models. They organize these papers following the structure of Algorithm, System, Application and Framework. When discussing these papers, the authors provide necessary background  and include tables and figures to make it easy for the readers to follow. This survey may be beneficial for readers working in related field.

**Audience:**

Yes

**Broader Impact Concerns:**

I don't see broader impact concerns.

**Claims And Evidence:**

Yes

**Requested Changes:**

1. The authors may want to include more comparisons of different strategies under the same aspect of efficiency (for example, in the Efficient Fine-tuning section) to make it clear for the readers to understand their advantages and disadvantages.

2. The overall structure of organizing the literatures need to change. As discussed in weakness 2, "Efficient Training" may not be a good aspect to discuss efficient diffusion models. A good reference would be the paper (Ma et al., 2024c) which mentioned by the authors in the introduction.

3. Application and framework could be more concise as they are not the focus of this survey. Moreover, in the abstract, when discussing application, the authors may want to include text and 3D.

4. Please fix the minor issues mentioned in the weakness.

**Strengths And Weaknesses:**

## Strengths:
1. This survey comprehensively reviews literatures about efficient diffusion, which shows the recent progress of efficient diffusion models.
2. This survey briefly introduces some background before discussing a topic which makes it easier for readers to get some basic knowledges.
3. Tables and figures are presented to explain some concepts and show the overall structure of the survey, which makes the survey clear to read.

## Weakness:
1. This survey does not include a good summarization or organization of papers listed.
    - This survey introduces many papers which reads like a related work in a paper without summarization or organization for the papers discussed. For instance, in section 3.2, the authors discussed LoRA, Adapter and ControlNet without a high level of discussion about their advantages and disadvantages when compared with each other. As a reader, it is hard to get a good feeling about which method I should  investigate based on my own application or issues at hand.

2. I have concerns about the discussion for "Efficient Training".
    - Most papers discussed in this section are about improving the quality of generation or improving the efficiency of inference, not about reducing cost of pre-training as mentioned in Section 3.1. For instance, as mentioned in the paper "The cosine noise schedule helps the model achieve high-quality sample generation by optimizing the noise distribution throughout the training process, allowing the model to generate samples with a reduced number of sampling steps while preserving essential structural information at each stage." Purpose of applying cosine noise schedule is to improve quality and reduce inference time, but no evidence or discussion shows it can improve the training efficiency.
    - The four categories discussed in this section are not at the same concept level. For instance, score matching and rectified flow are different formulation of variants. Noise scheduling and adaptive priors are generally training tricks which can be applied to different formulation of variants. To better organize this section, the authors may refer to (Ma et al. 2024c) in which they organize these papers as principles instead of discussing improve the training efficiency.
    - From my point of view, compare with applying diffusion model in pixel space, applying diffusion model in latent space is one way to really improve the training efficiency, however, there is no discussion about this part.

3. Questions about discussing application and Frameworks.
    - While application and frameworks are important, they are not the main focus of this survey. A short discussion about them may not good enough. Unless there are some novel techniques that are particularly designed for different applications or frameworks that cannot be included in the above discussion.
    - What is 'Training', 'Fine-Tuning' and 'inference' mean in table 2? Does it mean whether the frame work support these oprations?

4. Other issues:
    - Duplicate citation: \
         (a) "Adding conditional control to text-to-image diffusion models". Zhang et al 2023c and 2023d. \
         (b) " Diffusion models in vision: A survey", Croitoru 2023a and 2023b.
    - Typo:
        (a) However, this approach has been shown **ta** failure to leverage the linguistic features of the text in question (page 5).

---

> ### Author Response · Authors · 2025-03-23
> **Response to Reviewer TFC6**
>
> > Include more comparisons of different strategies under the same aspect of efficiency.
>
> Thanks for your valuable suggestion! **In response to this, we have expanded our discussion in Section 3.1, 3.2, 3.3 and their subsections to provide a more detailed comparison of various strategies.** **The revisions are highlighted in Sec. 3 in blue color.**
>
> > Improve the paper structure in Section 3.1.
>
> **(1) Is it worth having an “Efficient Training” category?**
>
> We thank the reviewer for this suggestion. **We agree that the original organization of this section conflated distinct concepts and lacked focus on pre-training efficiency. However, we argue that "Efficient Training" remains a critical and distinct category for diffusion models.** Training foundational diffusion models (e.g., Stable Diffusion) often requires weeks of GPU pre-training, with costs exceeding $600k for large-scale text-to-image models. Efficient training methods directly address this bottleneck by accelerating convergence. To clarify our focus, we have restructured this section with methods that improve training efficiency. **The revisions are highlighted in Sec. 3.1 in blue color.**
>
> **(2) New organization of literatures**
>
> **We have expanded the content by adding a section on "​Latent Diffusion" to better structure the literature**. As suggested, **we've created a new section called "Loss Formulation", which includes score matching and rectified flow**. Following your suggestion, ​**data-dependent adaptive priors and ​noise schedule have been categorized under the subsection ​training tricks, and we further incorporated ​extra loss supervision as a part of ​efficient training tricks**. These revisions aim to enhance clarity and align with your insightful suggestions. We hope these updates address your concerns effectively. **The revisions are highlighted in Sec. 3.1 in blue color.**
>
> **(3) Expand subsection "Latent Diffusion".**
>
> Thank you for your valuable and inspiring suggestions. **We have added a new subsection to discuss efficient methods that utilize diffusion models in latent space.** First, we introduce the development of diffusion models from pixel space to latent space in image-related tasks (image generation and editing), then we demonstrate other works of diffusion models in latent space that can efficiently complete other tasks, such as video generation, 3D generation, tabular data generation, RNA sequence generation. Please refer to the main text of modified submission for details. **The revisions are highlighted in Sec. 3.1.1 in blue color.**
>
> > Improve the paper structure in application and framework.
>
> Thanks for your valuable suggestions! **We have revised the framework section to be more concise**, focusing on frameworks with optimization techniques relevant to efficient diffusion (e.g., Flash Attention, xFormers, DeepSpeed, OneFlow for full training-to-inference support, and Stable-Fast, Onediff, DeepCache, TGATE, xDiT for inference acceleration). **We also clarified the "Training" and "Inference" terms in the table caption to explicitly indicate support for these operations.** Additionally, we have summarized the key features of each framework, highlighting their unique strengths and contributions to efficient diffusion, such as memory optimization, computational speed, or scalability, to provide a clearer comparison and better context for their applicability.
>
> > Other issues
>
> Thanks for your careful review! **We have corrected all the issues pointed out, as well as other errors we identified during revision.**

---

> > ### Comment · Reviewer_TFC6 · 2025-04-11
> >
> > Thanks for the response and thanks for being patient.
> >
> > The authors addressed most of my concerns. The overall structure is more clear to me know. One minor issue is that paper "Adding conditional control to text-to-image diffusion models" is still duplicated in the citation.

---

> > > ### Author Response · Authors · 2025-04-13
> > > **Response to Reviewer TFC6**
> > >
> > > >Paper "Adding conditional control to text-to-image diffusion models" is still duplicated in the citation.
> > >
> > > Thank you for your careful review! The duplicate citation of [1] in our manuscript has now been corrected in the revised version.
> > >
> > > [1] Zhang, Lvmin, Anyi Rao, and Maneesh Agrawala. "Adding conditional control to text-to-image diffusion models." Proceedings of the IEEE/CVF international conference on computer vision. 2023.

---

### Review · Reviewer_v2b1 · 2025-03-07

**Summary Of Contributions:**

This survey presents a systematic review of diffusion models with a focus on efficiency. It is a timely and popular topic in the research community.

**Audience:**

Yes

**Claims And Evidence:**

Yes

**Requested Changes:**

1. I believe that flow matching works should be highlighted in Section 2, in addition to the classical DDPM formulations, as they are increasingly demonstrating real-world impact. The authors have already cited the seminal works on flow matching (Lipman et al., 2022; Liu et al., 2022b; Albergo et al., 2023) and should expand on this. Placing it in Section 3.1.4, "Rectified Flow," in my opinion, unreasonably downplays its importance in the field.

2. Diffusion generative models for video are an important topic. While this paper focuses on surveying efficient methods, it is suggested that the authors review the latest developments in this field and expand the current single-paragraph Section 5.2, "Video Generation."  For example, consider referencing:
 **Melnik, A., Ljubljanac, M., Lu, C., Yan, Q., Ren, W., & Ritter, H.** *Video Diffusion Models: A Survey.* Transactions on Machine Learning Research.

3. Please use $\mathrm{d}$ (\mathrm{d}) to represent the differential operator, for example, in Equations (20)–(23).

4. The mathematical formulations of score-based models, SDEs, and probability flow ODEs should be introduced earlier in Section 2, "Fundamentals," to ensure that the mathematical concepts are more coherent and self-contained. After all, SDEs, ODEs, score-based models, and denoising diffusion can be understood from a very similar perspective. Presenting them together will make it clearer for readers unfamiliar with the field to grasp the connections and nuances.

5. More importantly, the survey should discuss and provide insights into potential future work, given the current research landscape on diffusion model efficiency. This will help readers not only understand the existing work, which is abundant, but also grasp key trends and research focus.

**Strengths And Weaknesses:**

1. Figures 1 and 2 clearly illustrate the paper's organization and are easy to follow. The writing is generally clear, at least for those familiar with diffusion models.
2. Methods from various application domains are surveyed and discussed.

---

> ### Author Response · Authors · 2025-03-23
> **Response to Reviewer v2b1**
>
> > Highlight the flow matching works in Section 2.
>
> Thanks for the suggestion! We agree that flow matching represents a crucial framework for diffusion models. **We add the detailed summarization of flow matching methods from Continuous Normalizing Flows (CNF) [1] to Conditional Flow Matching (CFM) [2] in Section 2.1.4 (Flow Matching).** Additionally, we highlight one particular flow matching approach called optimal transport to introduce the straight sampling trajectory of diffusion models in Section 3.1.4 (Rectified Flow). We appreciate your feedback on the importance of flow matching and have ensured it receives appropriate attention in our manuscript.
>
> > Expand the current Video Generation part.
>
> **We have carefully reviewed and added the comprehensive survey [3] that you recommended and have revised Section 2.2.2  (Video Generation) to better align with the latest developments.** Specifically, we incorporated insights and expanded the section by adding two works. [4] introduces cross-frame consistency constraints to enable efficient video editing via image diffusion models. [5] employs hierarchical diffusion strategies to enhance long video generation efficiency. Your guidance has significantly improved the depth and relevance of our survey, and we deeply appreciate your expertise in shaping this revision.
>
> > Please use d(\mathrm{d}) to represent the differential operator.
>
> **We have replaced all instances of ‘d’ with d(\mathrm{d}), which makes the formulas more correct in mathematical notation.**
>
> > The mathematical formulations of score-based models, SDEs, and probability flow ODEs should be introduced earlier in Section 2.
>
> **Following your advice, we have restructured Section 2.1 (Basic Formulas for Diffusion Models) to provide a more comprehensive and logical foundation.** We now present all fundamental concepts, including DDPM, score matching, solvers (SDE and ODE), and flow matching in a coherent sequence. This restructuring creates a more natural progression of ideas, helping readers better understand the mathematical connections between these different formulations and the underlying principles of diffusion models.
>
> > Discuss and provide insights into potential future work.
>
> **We expand Section 6 to include a forward-looking discussion on Future Work for efficient diffusion models.** This section now covers two key topics: Without Classifier-Free Guidance and Efficient Attention Mechanisms.
>
> [1] Ricky TQ Chen, Yulia Rubanova, Jesse Bettencourt, and David K Duvenaud. Neural ordinary differential equations. Advances in neural information processing systems, 31, 2018.
>
> [2] Yaron Lipman, Ricky TQ Chen, Heli Ben-Hamu, Maximilian Nickel, and Matt Le. Flow matching for generative modeling. arXiv preprint arXiv:2210.02747, 2022.
>
> [3] Melnik, Andrew, et al. "Video diffusion models: A survey." arXiv preprint arXiv:2405.03150 (2024).
>
> [4] Duygu Ceylan, Chun-Hao Paul Huang, and Niloy J Mitra. Pix2video: Video editing using image diffusion. arXiv preprint arXiv:2303.12688, 2023.
>
> [5] Shengming Yin, Chenfei Wu, Huan Yang, Jianfeng Wang, Xiaodong Wang, Minheng Ni, Zhengyuan Yang, Linjie Li, Shuguang Liu, Fan Yang, et al. Nuwa-xl: Diffusion over diffusion for extremely long video generation. arXiv preprint arXiv:2303.12346, 2023.

---

> > ### Comment · Reviewer_v2b1 · 2025-03-23
> >
> > Thank you for your response. My concerns above have been largely addressed. However, Section 6 (Future Work) feels rather brief, which makes the overall structure of the article appear unbalanced—for example, Sections 3 and 4 are quite long and contain many subsections, while Sections 5 and 6 are noticeably shorter and more abrupt in comparison. Additionally, the discussion on efficient attention seems too general, as it could apply to any neural network architecture, regardless of whether diffusion models are used.
> >
> > Also, I'd like to add a few comments regarding some recent works on efficient diffusion that I’ve come across. The fast diffusion approach introduced in Figure 6 for motion prediction has also been further studied in recent work through distillation by a generative model [1].
> > In the context of behavior modeling, recent studies on scene simulation via diffusion [2] have adopted amortized diffusion to address spatial-temporal data generation. These two works may help complement the discussion on the broader topic of autonomous driving and robotic navigation.
> >
> > Additionally, I’d like to draw your attention to recent efforts on consistency modeling for 3D data generation [3–4], which may also be relevant to the ongoing discussion.
> >
> >
> > [1] Fu Y, Yan Q, Wang L, Li K, Liao R. MoFlow: One-Step Flow Matching for Human Trajectory Forecasting via Implicit Maximum Likelihood Estimation based Distillation. arXiv preprint arXiv:2503.09950. 2025 Mar 13.
> >
> > [2] Jiang M, Bai Y, Cornman A, Davis C, Huang X, Jeon H, Kulshrestha S, Lambert J, Li S, Zhou X, Fuertes C. Scenediffuser: Efficient and controllable driving simulation initialization and rollout. Advances in Neural Information Processing Systems. 2024 Dec 16;37:55729-60.
> >
> > [3] Du BA, Hu W, Liao R. Multi-scale Latent Point Consistency Models for 3D Shape Generation. arXiv preprint arXiv:2412.19413. 2024 Dec 27.
> >
> > [4] Wu Z, Zhou P, Yi X, Yuan X, Zhang H. Consistent3d: Towards consistent high-fidelity text-to-3d generation with deterministic sampling prior. InProceedings of the IEEE/CVF Conference on Computer Vision and Pattern Recognition 2024 (pp. 9892-9902).

---

> > > ### Author Response · Authors · 2025-03-31
> > > **Response to Reviewer v2b1**
> > >
> > > > A longer discussion on areas of future work, and the discussion on efficient attention seems too general.
> > >
> > > Thanks for your valuable suggestions! **We have expanded the Future Work section with additional details and broader perspectives, and added a new subsection titled "Hybridizing Diffusion and Autoregressive Models"**. Additionally, **we refined the discussion on efficient attention mechanisms to make it more specific to diffusion models**. We believe these additions significantly strengthen the paper's forward-looking contributions.
> > >
> > > > Recent works on efficient diffusion.
> > >
> > > Thank you for your valuable suggestions. **We’ve added the latest work by [1] as recommended in Section 3.1.3. We’ve included SceneDiffuser[2] in the same paragraph**, noting its use of amortized diffusion to optimize efficiency for driving simulation, complementing the discussion on adaptive priors and training efficiency.
> > >
> > > > Recent efforts on consistency modeling for 3D data generation.
> > >
> > > Thanks for pointing out this! **We have carefully reviewed the paper you proposed. Now the revised manuscript has added the [3-4] in Section 3.3.1 and Section 3.3.4.** [3] uses ODE's deterministic trajectory, combined with fixed noise and CDS loss, which minimizes stochastic fluctuations and computational redundancy. [4] implements one-step generation through consistency distillation, combines multi-scale latent space and 3D attention mechanisms to reduce computational complexity.
> > >
> > > [1] Fu Y, Yan Q, Wang L, Li K, Liao R. MoFlow: One-Step Flow Matching for Human Trajectory Forecasting via Implicit Maximum Likelihood Estimation based Distillation. arXiv preprint arXiv:2503.09950. 2025 Mar 13.
> > >
> > > [2] Jiang M, Bai Y, Cornman A, Davis C, Huang X, Jeon H, Kulshrestha S, Lambert J, Li S, Zhou X, Fuertes C. Scenediffuser: Efficient and controllable driving simulation initialization and rollout. Advances in Neural Information Processing Systems. 2024 Dec 16;37:55729-60.
> > >
> > > [3] Du BA, Hu W, Liao R. Multi-scale Latent Point Consistency Models for 3D Shape Generation. arXiv preprint arXiv:2412.19413. 2024 Dec 27.
> > >
> > > [4] Wu Z, Zhou P, Yi X, Yuan X, Zhang H. Consistent3d: Towards consistent high-fidelity text-to-3d generation with deterministic sampling prior. InProceedings of the IEEE/CVF Conference on Computer Vision and Pattern Recognition 2024 (pp. 9892-9902).

---

> > > > ### Comment · Reviewer_v2b1 · 2025-03-31
> > > >
> > > > Thanks for your responses. I don’t have any more concerns as of now.

---

### Review · Reviewer_AZ4W · 2025-03-09

**Summary Of Contributions:**

The authors provide a survey on *efficient* diffusion models. They cover a broad range of research, arranging it into the categories of

**Audience:**

Yes

**Claims And Evidence:**

Yes

**Requested Changes:**

See weaknesses above. I would classify none of them as critical, as I believe the paper to be satisfactory as a survey. However, I hope the authors consider my feedback; I especially believe 1 and 2 would improve the experience for the reader.

**Strengths And Weaknesses:**

**Strengths**
- The paper is clearly structured and easy to read, with different research tidily organised into their corresponding sections.
- A wide range of topics is covered, spanning algorithms and systems, giving the reader a clear picture of efficient diffusion *across the whole stack*.
- There is generally a good balance between detail and brevity.


**Weaknesses**
1. I find that the authors dive into the details too quickly, when more general background/motivation/intuition for each section would help. For example, Sec. 3.4.1 would benefit from initially discussing the *benefits* of low-bitwidth and fixed-precision inference, such as reduced memory costs and smaller-area/lower-power hardware at the start. Similarly, the first paragraph of Sec. 2 could have a sentence saying that diffusion models generate data by iterative denoising, transforming pure noise gradually into clean data. For a final example, the opening of Sec. 3.3 could include some elaboration on how the iterative denoising process is slowed down by the need to *sequentially* perform a large number of network inferences. **The paper would benefit from simply one or two additional sentences at the start of each section/subsection to help the reader in this way.**
2. The discussion of diffusion model fundamentals relies on the original DDPM paper which complicates understanding when later discussion of SDE and ODE samplers pops up. I would suggest using a more unified mathematical framework such as the one in https://diffusionflow.github.io/.
3. Distillation would be better placed in efficient sampling, since its purpose is not to reduce network size, but to reduce the number of sampling steps. Also, rectified flows would better be considered together with distillation. Finally, consistency models are not very well discussed (LCM-LoRA is introduced before CMs themselves).
4. Links/references should be provided to the frameworks discussed in Sec. 6. Additionally it is not completely clear what a "framework" is in this context.
5. Miscellaneous missing references:
    - Distillation: Yin et al. One-step Diffusion with Distribution Matching Distillation, CVPR 2024
    - Caching: Shen et al. MD-DiT: Step-aware Mixture-of-Depths for Efficient Diffusion Transformers, AFM Workshop NeurIPS 2024; $\Delta$-DiT: Accelerating Diffusion Transformers without training via Denoising Property Alignment-DiT: Accelerating Diffusion Transformers without training via Denoising Property Alignment, Chen et al. 2024
    - Parallel Inference: Kodaira et al. StreamDiffusion: A Pipeline-level Solution for Real-time Interactive Generation,  2023
    - Efficient Sampling: Xu et al. Restart Sampling for Improving Generative Processes,  NeurIPS 2023
6. **Edit (additional point)** There is a lack of discussion about future work; I think the paper would benefit from identifying to the reader broader areas of research potential, especially in applications beyond image generation.

---

> ### Author Response · Authors · 2025-03-23
> **Response to Reviewer AZ4W**
>
> > More general background/motivation/intuition for each section.
>
> **In response to this, we have expanded our discussion in Section 3.1, 3.2, 3.3 and their subsections to provide more general background/motivation/intuition. In Section 2, we have restructured the content and added a concise summarization of DDPM**, providing a general definition that explains how diffusion models leverage forward and backward stochastic processes to gradually transform data into noise and then reverse this process for generation.
>
> > More unified mathematical framework.
>
> Thanks for the suggestion! **After reviewing the recommended resource, we have restructured Section 2.1 (Basic Formulas for Diffusion Models) to present a more cohesive mathematical framework.** The revised section now provides a comprehensive overview that systematically introduces all relevant mathematical frameworks including DDPM, score matching, various solvers (SDE, ODE), and flow matching in a logical progression. We have also enhanced the comparative analysis between these methods to better illustrate their interconnections and unique properties. We believe these changes will provide readers with a more intuitive understanding of the relationships between different diffusion formulations while making the theoretical foundations more accessible.
>
> > Categorization issues.
>
> **We agree that distillation should be placed within the "Efficient Sampling" section, as its primary purpose is to reduce sampling steps rather than network size.** We have implemented this change in our revised manuscript.
> Regarding rectified flows, after careful consideration, we have maintained their placement in the "Efficient Training" section rather than grouping them with distillation. While rectified flows can indeed be utilized in distillation processes, our focus in this survey is on highlighting their significant contributions to training efficiency and their capability for straight-through sampling.
>
> > Expand the discussion on Consistency Models.
>
> **We have carefully reviewed the timeline of the CMs development.** Now we discuss CMs through two technical lenses. In Section 3.3.1(Efficient Solvers) we analyze their theoretical connections to diffusion frameworks through advanced ODE discretization techniques. In Section 3.3.4(Vector Field Distillation) we examine CMs' proceed innovation in probability flow distillation and one-step generation capabilities.
>
> > Provide links/references for frameworks and clarify the definition of "framework".
>
> **We have added links/references to the frameworks in Section 5. Additionally, we have clarified the definition in Section 5.**  We define a "framework" here as a specialized software system or library that provides optimized tools and pipelines specifically for accelerating training, inference, or both in diffusion models, such as xFormers for attention optimization or DeepSpeed for distributed training.
>
> > Miscellaneous missing references.
>
> **We have carefully reviewed the papers you mentioned and added the papers into our revised manuscript.**
>
> > Discuss and provide insights into potential future work.
>
> **We expand Section 6 to include a forward-looking discussion on Future Work for efficient diffusion models. This section now covers two key topics: Without Classifier-Free Guidance and Efficient Attention Mechanisms.**

---

> > ### Comment · Reviewer_AZ4W · 2025-03-23
> >
> > Thanks for the update, it seems like the paper has improved considerably. I will get back to the authors if I have any further comments after reading the updated manuscript during the week.

---

> > > ### Comment · Reviewer_AZ4W · 2025-03-26
> > >
> > > After going over the paper, I am satisfied with the manuscript. However, I will still leave some additional feedback to the authors.
> > > 1. A *longer discussion* on areas of future work, which the authors deem to have high potential, would really benefit readers looking to enter/explore the field of research.
> > > 2. Section 2.1 would benefit from more explicit connections between the *equivalencies* of different formulations of diffusion (again see https://diffusionflow.github.io/).

---

> > > > ### Author Response · Authors · 2025-03-31
> > > > **Response to Reviewer AZ4W**
> > > >
> > > > > A longer discussion on areas of future work.
> > > >
> > > > Thanks for your valuable suggestions! **We have expanded the Future Work section with additional details and broader perspectives, and added a new subsection titled "Hybridizing Diffusion and Autoregressive Models"**. Additionally, **we refined the discussion on efficient attention mechanisms to make it more specific to diffusion models**. We believe these additions significantly strengthen the paper's forward-looking contributions.
> > > >
> > > >
> > > > > Section 2.1 would benefit from more explicit connections between the equivalencies of different formulations of diffusion (again see https://diffusionflow.github.io/)
> > > >
> > > > Thanks for your valuable suggestion! **We have reviewed this link again and added more detailed summarization about the equivalence between DDPM, score-based models, and flow matching**. **We have also included a description of how we use flow matching in diffusion-like optimal transport in Section 3.1.2. Furthermore, in Section 2, we have added connections to each technology to provide a clearer explanation of the differences within context.**

---

### Author Response · Authors · 2025-03-23
**General Response**

We would like to express our sincere gratitude to all the reviewers for their time and constructive feedback. We have revised our manuscript and submitted an updated version for review. **The revisions are highlighted in blue color.**

## Summary of Revisions

### Section 2
1. We have merged the original Section 5: Application and Section 2: Fundamentals of Diffusion Models into the new Section 2: Background and Applications.
2. We add and highlight flow matching methods from Continuous Normalizing Flows (CNF) [1] to Conditional Flow Matching (CFM) [2] in Section 2.1.4 (Flow Matching). (Reviewer v2b1)
3. We have replaced all instances of ‘d’ with d(\mathrm{d}). (Reviewer v2b1)
4. We have restructured Section 2.1 (Basic Formulas for Diffusion Models) to provide a more comprehensive and logical foundation. We now present all fundamental concepts, including DDPM, score matching, solvers (SDE and ODE), and flow matching in a coherent sequence. (Reviewer v2b1, Reviewer AZ4W)
5. We carefully reviewed the comprehensive survey [3] and have revised Section 2.2.2  (Video Generation) to better align with the latest developments. (Reviewer v2b1)

### Section 3
1. We expand our discussion in Section 3.1, 3.2, 3.3 and their subsections to provide a more detailed comparison of various strategies. (Reviewer TFC6, Reviewer AZ4W)
2. We have added a new subsection to discuss efficient methods that utilize diffusion models in latent space. (Reviewer TFC6)
3. We restructured Section 3.1 (Efficient Training) with methods that improve training efficiency (Reviewer TFC6), including:
	1. Latent Diffusion
	2. Loss Formulation
	3. Training Tricks
4. We placed distillation within Section 3.3 "Efficient Sampling", as its primary purpose is to reduce sampling steps rather than network size. (Reviewer AZ4W)
5. We expand the discussion on Consistency Models. (Reviewer AZ4W)

### Section 4
1. We add missing references. (Reviewer AZ4W)
2. We expand our discussion to provide a detailed summary of various strategies. (Reviewer TFC6)

### Section 5
1. We have revised the framework section to be more concise, focusing on frameworks with optimization techniques relevant to efficient diffusion. (Reviewer TFC6)
2. We provide links/references for frameworks and clarify the definition of "framework". (Reviewer AZ4W)

### Section 6
1. We add Section 6 to include a forward-looking discussion on Future Work(Reviewer v2b1, Reviewer AZ4W), including:
	1. Without Classifier-Free Guidance;
	2. Efficient Attention Mechanisms.

[1] Ricky TQ Chen, Yulia Rubanova, Jesse Bettencourt, and David K Duvenaud. Neural ordinary differential equations. Advances in neural information processing systems, 31, 2018.

[2] Yaron Lipman, Ricky TQ Chen, Heli Ben-Hamu, Maximilian Nickel, and Matt Le. Flow matching for generative modeling. arXiv preprint arXiv:2210.02747, 2022.

[3] Melnik, Andrew, et al. "Video diffusion models: A survey." arXiv preprint arXiv:2405.03150 (2024).

---

### Decision · Action_Editor_euqR · 2025-04-29

**Recommendation:** Accept as is

**Comment:**

Please continune to update the literature in this field on the github page mentioned in this paper.

**Audience:**

This survey paper is of interest to researchers and practitioners who use diffusion models.

**Claims And Evidence:**

This paper reviews the recent developments of efficient diffusion models. Overall, it is well organized and the content is useful to practitioners. The discussed topics are of interest to the TMLR's audience.